# TOF<sub>IMS</sub> mass spectrometry-based immunopeptidomics refines tumor antigen identification

Naomi Hoenisch Gravel [1,2,3], Annika Nelde [1,2,3], Jens Bauer [1,2,3], Lena Mühlenbruch[1,2,3,4], Sarah M. Schroeder [1,2,5], Marian C. Neidert[6,7,8], Jonas Scheid [1,2,3,9], Steffen Lemke[1,2,3,9], Marissa L. Dubbelaar [1,2,3,9], Marcel Wacker [1,2,3], Anna Dengler[1,2,3], Reinhild Klein [10], Paul-Stefan Mauz[5], Hubert Löwenheim[5], Mathias Hauri-Hohl[11], Roland Martin [12], Jörg Hennenlotter [13], Arnulf Stenzl[13], Jonas S. Heitmann [3,15], Helmut R. Salih [3,4,14], Hans-Georg Rammensee[2,3,4] & Juliane S. Walz [1,2,3,14] ✉

T cell recognition of human leukocyte antigen (HLA)-presented tumor-associated peptides is central for cancer immune surveillance. Mass spectrometry (MS)-based immunopeptidomics represents the only unbiased method for the direct identification and characterization of naturally presented tumor-associated peptides, a key prerequisite for the development of T cell-based immunotherapies. This study reports on the implementation of ion mobility separation-based time-of-flight (TOF<sub>IMS</sub>) MS for next-generation immunopeptidomics, enabling high-speed and sensitive detection of HLA-presented peptides. Applying TOF<sub>IMS</sub>-based immunopeptidomics, a novel extensive benign<sub>TOFIMS</sub> dataset was generated from 94 primary benign samples of solid tissue and hematological origin, which enabled the expansion of benign reference immunopeptidome databases with >150,000 HLA-presented peptides, the refinement of previously described tumor antigens, as well as the identification of frequently presented self antigens and not yet described tumor antigens comprising low abundant mutation-derived neoepitopes that might serve as targets for future cancer immunotherapy development.

T cell recognition of human leukocyte antigen (HLA)-presented peptides plays a key role in the immune surveillance of malignant diseases[1,2]. Various T cell-based immunotherapeutic approaches aim to utilize respective tumor antigens to therapeutically induce anti-tumor T cell responses[3–6]. Thus, identifying suitable antigen targets that show natural, high frequent, and tumor-exclusive presentation on the tumor cell surface and are recognized by the immune system is central for the success of these therapeutic approaches[7]. With regard to tumor-exclusive presentation of HLA-presented peptides not derived from tumor-specific mutations, knowledge on respective peptides presentated on benign tissues is of key importance[8,9]. However the availability of reference databases of such benign tissue derived HLA ligands remains limited. Mass spectrometry (MS)-based immunopeptidomics represents the only unbiased method to identify and characterize such naturally presented HLA class I- and HLA class II-restricted peptides on the cell surface[10,11].

Despite immense technical improvements and optimized immunopeptidomics workflows in the last decades[12,13], sensitivity of shotgun MS discovery approaches remains limited and so far cannot capture the entirety of the immunopeptidome, which represents a highly dynamic, rich, and complex assembly of peptides. Moreover, the MS-based identification of HLA-presented peptides is further hindered by

the low abundance, in particular described for mutation-derived neoepitopes, the distinct length, and the specific physicochemical properties of HLA-presented peptides as compared to standard proteomics using tryptic digests[11,12,14].

On-line coupled ion mobility separation (IMS) coupled MS-technology (e.g., trapped ion mobility separation (TIMS) and high field asymmetric waveform ion mobility spectrometry (FAIMS)), providing an additional and orthogonal separation dimension the so-called collisional cross section (CCS), was suggested as next-generation tool increasing sensitivity and high-speed analysis of large cohort samples[15–17]. In addition to the standard parameters retention time (RT), mass to charge ratio, and fragment spectra, multiple ion mobility resolved MS scans ensure high-resolution MS analysis. This technology has been successfully implemented in other MS-based *omics* areas, comprising proteomics, metabolomics or lipidomics, showing increased identifications based on faster tandem (MS/MS) scan rates and acquisition without a loss of sensitivity[18–20]. In contrast, IMS-coupled MS method implementation and large-scale application in immunopeptidomics is so far limited[12,21–25] and using MS such as Orbitrap without IMS technology still reflects the current state-of-the-field[8,26–28].

Here we report on the implementation of TOF$_{IMS}$ MS for immunopeptidomics and its application for next-generation tumor antigen discovery. TOF$_{IMS}$-based immunopeptidomics enabled (i) the large-scale expansion of benign reference databases providing novel insights in the immunopeptidome landscape and the refinement of non-mutated tumor-associated antigen (TAA) definition and (ii) the de novo discovery of not yet described tumor antigens, comprising frequently presented self antigens as well as low abundant mutation-derived neoepitopes, as potential targets for cancer immunotherapy.

## Results
### TOF$_{IMS}$ MS enables large-scale identification of naturally HLA-presented peptides

To establish a method for immunopeptidomics using TOF$_{IMS}$ MS, the monoallelic EBV-transformed human B cell line JY (HLA-A*02, HLA-B*07, HLA-C*07) was analyzed in two dilutions (high peptide concentration (JY 1) and low peptide concentration (JY 2)). First, the liquid chromatography (LC) workflow was optimized and validated, evaluating two different gradient types and four different gradient lengths (Fig. 1a, Supplementary Fig. 1a, Supplementary Data 1). For all gradient types and lengths identified HLA class I-presented peptides showed similar hydropathy profiles (Supplementary Fig. 1b) and length distribution, as expected for HLA class I-presented peptides (Supplementary Fig. 1c). Highest number of HLA class I-presented peptide identifications (1895 for JY 1 and 1252 for JY 2) were detected with the gradient type A with a length of 60 min, which was used for the further implementation of the subsequent MS method. For MS optimization, the impact of various parameters on peptide yields was evaluated (Fig. 1b, Supplementary Data 1), reaching HLA class I peptide yields of 5691 for JY 2 and resulting in a final method for TOF$_{IMS}$-based immunopeptidomics (Table 1).

Analyzing the CCS of identified peptide spectrum matches (PSMs) showed that TOF$_{IMS}$'s technology separated ions with similar RT orthogonally according to their CCS values, enabling identification of co-eluting peptides and thus a higher sensitivity (Fig. 1c, d). The identified peptides showed an expected mass distribution from 700 to 1700 Da with the majority around 1000 Da for HLA class I-presented peptides (Fig. 1e). HLA class II peptides showed an expected mass distribution ranging from 700 to 2600 Da with the majority of peptides slightly heavier than HLA class I-presented peptides with 1400–1800 Da (Fig. 1f). Identified HLA class I-presented peptides showed the typical length distribution with 9-mers making up 70% of peptides (Fig. 1g). HLA class II-presented peptides showed a typical length distribution with the majority at about 15 amino acids (Fig. 1h).

To further delineate the eligibility of TOF$_{IMS}$ for HLA-presented peptide analysis, in particular to provide novel insights to and expansion of the immunopeptidome landscape of large-scale benign and malignant datasets, an alignment of primary benign and malignant samples (*n* = 10, Supplementary Data 2) analyzed using TOF$_{IMS}$ and the current state-of-the-field (Orbitrap) technology, applied in current immunopeptidome references[26–28], was performed. Peptide length and mass distribution, quality score (−10lgP), frequency of HLA allotype allocation, and technical overlap were comparable for the two technologies (Supplementary Fig. 2a–h). Low-frequent peptide artefacts resulting from proteolytic fragmentation originating from endogenous peptidases[29] were identified with a median of 2.0% (range 0.1–6.5%) by TOF$_{IMS}$ and 0.5% (range 0.0–7.2%) by Orbitrap (Supplementary Fig. 2i). Of note, 91% of peptides classified as proteolytic were not annotated as HLA ligands. Focussing on the normalized reported area per peptide, 41% of TOF$_{IMS}$ and 20% of Orbitrap were exclusively identified in the lower rank (Supplementary Fig. 2j).

A median of 89% and 96% of all identified HLA class I ligands and HLA class II-presented peptides could be identified by TOF$_{IMS}$, respectively. Up to 57% of HLA class I ligands and 76% of HLA class II-presented peptides exclusive for TOF$_{IMS}$ datasets, which were significantly more hydrophobic and identified during the whole LC separation run (Supplementary Fig. 2k–o).

### TOF$_{IMS}$-based immunopeptidomics application for benign tissue-derived dataset

The relevance of benign immunopeptidome databases as reference has widely been recognized in the search for immunotherapy-relevant tumor-associated antigen (TAA) discovery. Based on the increase in HLA-presented peptide discovery using TOF$_{IMS}$ MS, HLA class I and HLA class II immunopeptidome analyses of benign primary samples (*n* = 92 for HLA class I, *n* = 94 for HLA class II), comprising solid tissues of 28 different organ origins and peripheral blood mononuclear cell (PBMC) samples, were performed (benign$_{TOFIMS}$ dataset, Fig. 2a, Supplementary Data 2). The HLA allotypes included within the benign dataset represent 99% of the world population with at least one allotype (Fig. 2b, c). A median of 3720 (range 41–15,139) HLA class I ligands and 5062 (range 125–14,330) HLA class II-presented peptides were identified across the samples (*n* = 92 for HLA class I, *n* = 94 for HLA class II, Fig. 2d, Supplementary Data 3). In total, 137,463 unique HLA class I ligands and 175,469 HLA class II-presented peptides were identified. As described before[8], the samples showed tissue-dependent peptide yields (Fig. 2e). Up to 63% (median 54%, range 32–63%) and 64% (median 54%, range 35–64%) of HLA class I and HLA class II-presented peptides, respectively, were identified in all technical replicates of a sample (Fig. 2f). The identified peptides showed the common length distribution for HLA class I- and HLA class II-presented peptides (Fig. 2g). Donor-specific samples derived from different tissues showed a heterogenous peptide overlap between the tissue samples (Fig. 2h, i). For HLA class I, up to 50% (mean 46%, range 42–50%) of all HLA class I peptides were identified in only one sample from the same donor, 0.5% (mean 0.2%, range < 0.1–0.5%) were found in all tissues of one donor. For HLA class II-presented peptides up to 73% (mean 69%, range 66–73%) and 0.1% (mean 0.1%, range 0.1–0.1%) of HLA class II-presented peptides were found in one and all tissues of one donor, respectively.

### Benign$_{TOFIMS}$ dataset expands benign references of HLA-presented peptides

Comparing the generated benign$_{TOFIMS}$ immunopeptidome dataset with published benign immunopeptidome repositories[8,30,31], 46% of HLA class I ligands and 54% of HLA class II-presented peptides have not yet been described in either the Immune Epitope Database IEDB[31] (*n* = n.a.) or other benign datasets acquired using state-of-the-field Orbitrap technology[8,30] (*n* = 297, Fig. 3a, b). Despite containing only

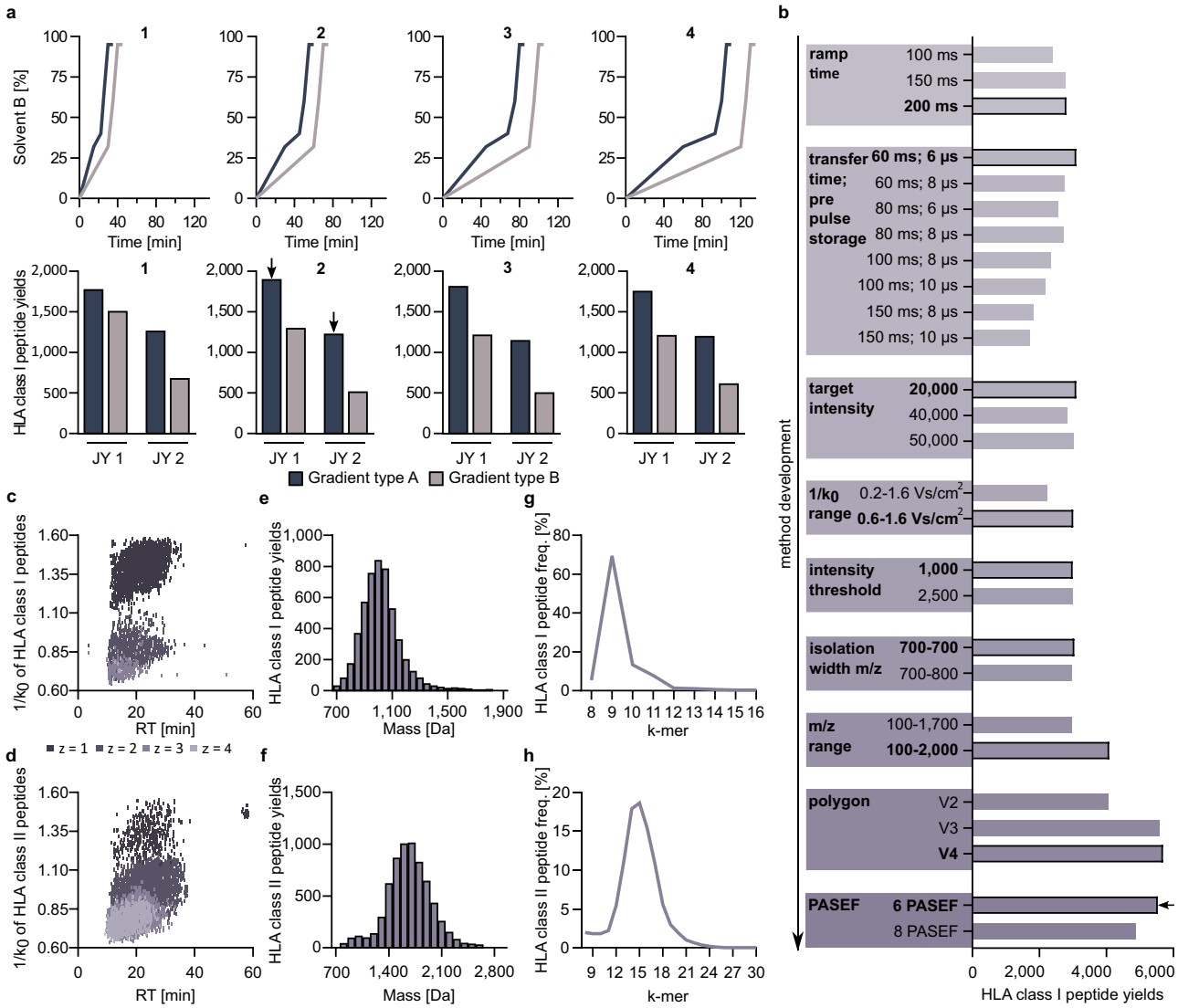

**Fig. 1 | Method development for immunopeptidomics using liquid chromatography-coupled TOF$_{IMS}$ MS. a** Different liquid chromatography gradient lengths (1, 2, 3, and 4) and gradient types (gradient A and B, upper row) with corresponding HLA class I-presented peptide yields for different JY human cell line sample concentrations (JY 1 and JY 2, lower row). Selected methods are indicated with arrows. **b** MS method development investigating different parameters with corresponding HLA class I peptide yields in JY 1 samples. Optimal parameters are bordered and the finally selected method (implemented in all further experiments) is indicated with an arrow, see Table 1 for more details. **c**, **d** Distribution of the collisional cross section (1/k$_0$) of PSMs of HLA class I- (**c**) and II-presented peptides (**d**) and in a JY 2 sample across the retention time and different charge states (z). **e**, **f** Histogram of peptide mass distribution of HLA class I- (**e**) and HLA class II-presented peptides (**f**) identified in a JY 2 sample. **g**, **h** Length distribution of identified HLA class I- (**g**) and HLA class II-presented peptides (**h**). HLA Human leukocyte antigen, min minutes, Da Dalton, PSM Peptide spectrum match, RT retention time, freq Frequency. Source data are provided as a Source Data file.

one-third of samples ($n = 92$ for HLA class I, $n = 94$ for HLA class II), 1.5-fold the amount of peptides were identified in the TOF$_{IMS}$ benign dataset, showing a mean immunopeptidome size of > 4700 compared to > 1500 peptides per sample for the other published benign Orbitrap datasets ($n = 297$, Fig. 3c). Benign$_{TOFIMS}$-exclusive peptides were significantly more hydrophobic, with a median GRAVY score of 0.13 for HLA class I ligands (range −3.50 to 4.03) and −0.26 for HLA class II-presented peptides (range −3.54 to 4.33) compared to previously described benign-associated peptides (median 0.133 and −0.45, range −3.77 to 4.40 and −4.04 to 3.87, respectively, Fig. 3d). This results in a median predicted immunogenicity of 0.05 (range −0.76 to 0.78) for the exclusive benign$_{TOFIMS}$ HLA class I ligands compared to 0.02 (range −0.86 to 0.66) for previously described benign peptides (Fig. 3e). The benign$_{TOFIMS}$-exclusive peptides do not differ from the published benign-associated peptides[8,30,31] in terms of amino acid composition for both HLA class I ligands (Fig. 3f) and HLA class II-presented

peptides (Fig. 3g). For biological relevance the source proteins were annotated to KEGG pathways and their subcellular location according to the human protein atlas. In both cases, benign$_{TOFIMS}$ exclusive peptides did not differ from published benign associated peptides (Fig. 3h–k)

### Benign$_{TOFIMS}$ immunopeptidome dataset refines the identification of tumor antigens for peptide-based immunotherapy

Comparative immunopeptidome profiling is central for the selection of tumor antigens to be applied in immunotherapeutic approaches. Tumor-exclusive presentation without representation of the respective antigen on benign tissue enables tumor-directed immune targeting without the risk of on-target-off-tumor adverse events. The novel benign$_{TOFIMS}$ immunopeptidome dataset rejected between 28% and 60% (median 40%) of previously published TAAs for various malignant diseases (ovarian carcinoma (OvCa)[32], chronic lymphocytic leukemia

**Table. 1 | TOF$_{IMS}$ method optimized for immunopeptidomics**

| | MS parameter | Setting |
|---|---|---|
| source | Source | captive spray |
| | Capillary | 1500 V |
| TIMS settings | 1/k$_0$ start to end (locked) | 0.6–1.6 Vs/cm$^2$ |
| | Ramp time | 200 ms |
| | Accumulation time | 200 ms |
| | Duty cycle (locked) | 100% |
| | Ramp rate | 4.85 Hz |
| tune | Collision energy | 10 eV |
| | Collision RF | 1500 Vpp |
| | Transfer time | 60 µs |
| | Pre pulse storage | 6 µs |
| | Mass spectra peak detection | |
| | Absolute threshold | 10 |
| | Absolute threshold (per 100 ms accu time) | 5 |
| | PASEF data | |
| | Denoising mode | No reduction |
| | Mobility peak detection | |
| | Absolute threshold | 5000 |
| MS/MS | Number of PASEF ramps | 6 |
| | Total cycle time | 1.44 s |
| | Charge | 0–5 |
| | Scheduling | Linear |
| | Precursor repetition | |
| | Target intensity | 20,000 |
| | Intensity threshold | 1000 |
| | Active exclusion | Activated |
| | Release after | 0.4 min |
| | Collision energy settings | |
| | 1/k$_0$ | 0.6–1.6 Vs/cm$^2$ |
| | | 20–59 eV |
| MS | m/z | 100–2000 |
| | Polarity | Positive |
| | Scan mode | PASEF |

*eV* Electronvolt, *Vpp* Voltage peak-to-peak, *min* Minutes, *MS* Mass spectrometer, *PASEF* Parallel accumulation-serial fragmentation, *accu* Accumulation, *m/z* mass to charge ratio.

(CLL)[33] and chronic myeloid leukemia (CML)[30] as they are not tumor-exclusive anymore (Fig. 4a, b). Whereas the OvCa rejected peptides were identified in samples of multiple benign tissue origin, >45% of rejected CLL and CML peptides were identified in samples of hematological origin (Supplementary Fig. 3a) within the benign$_{TOFIMS}$.

To detect previously undescribed TAAs using TOF$_{IMS}$ MS-based immunopeptidomics, we performed comparative immunopeptidome analyses of primary malignant samples (n = 2, RCC, HNSCC; Supplementary Data 2) with the TOF$_{IMS}$ and published benign immunopeptidome datasets[8,30]. Of the 13,517 total identified HLA class I ligands from RCC 89% (12,054/ 13,517) were identified using TOF$_{IMS}$ MS and 43% (5863/ 13,517) using Orbitrap MS. 30% (3999/13,517) of identified HLA class I ligands were found to be tumor-exclusive, of which 77% (3069/3999) were exclusively identified using TOF$_{IMS}$ (Fig. 4c, d). Similar observations were made for the HNSCC sample (2281 tumor-exclusive HLA class I ligands, of which 66% were TOF$_{IMS}$-exclusive; Supplementary Fig. 3b). For HLA class II-presented peptides, 8166 unique peptides were identified on the RCC tumor sample, of which 96% (7873/8166) and 42% (3389/8166) were identified by TOF$_{IMS}$. 33% (2676/8166) of identified peptides were tumor-exclusive, of which 72%

(1943/ 2676) were identified via TOF$_{IMS}$ (Fig. 4c). Similar observations were made for the HNSCC sample (6133 tumor-exclusive peptides, of which 60% were TOF$_{IMS}$-exclusive; Supplementary Fig. 3c).

As an exemplary use of the benign$_{TOFIMS}$ dataset, HLA class I and class II immunopeptidome profiling using TOF$_{IMS}$ MS was performed from primary CLL samples (n = 22, Supplemtary Data 2). The HLA alloytpes include in the CLL cohort were comparable to the alloytpes included in published datasets as well as the benign$_{TOFIMS}$ dataset (Supplementary Fig 4a–c). A median of 11,706 HLA class I ligands (range 5976 to 18,115) and 7833 HLA class II-presented peptides (range 2208 to 10,484) were identified per sample (Supplementary Fig 3d and Supplementary Data 3). In total 121,871 unique HLA class I ligands and 86,785 HLA class II-presented peptides were identified from the total sample cohort. Comparative immunopeptidome profiling was performed with published benign datasets[8,34] and resulted in 68% of HLA class I ligands (83,074 peptides, Fig. 4e) and 72% of identified HLA class II-presented peptides (62,224 peptides, Fig. 4f) to remain CLL-exclusive. A further alignment of these CLL-exclusive HLA class I and class II peptides using the novel benign$_{TOFIMS}$ dataset revealed (Fig. 4 e, f). 46% of CLL HLA class I ligands (55,812 peptides) and 68% of CLL HLA class II peptides (49,147 peptides) CLL-exclusive, thus annotating an additional 27,262 HLA class I ligands and 13,077 HLA class II-presented peptides as benign compared to previously published data (Supplementary Fig 4e, f). 727 HLA class I and 1556 HLA class II CLL-exclusive peptides showed a broard presentation within the CLL cohort with a frequency above 20% (up to 59% for HLA class I and 77% for HLA class II). Allotype specific peptide alignment of the most abundant HLA allotypes within the cohort (HLA-A*02, HLA-B*35 and HLA-C*07, Supplemtary Fig. 4g–i) even revealed CLL-exclusive peptides with presentation in up to 100% of HLA-matching samples (HLA-A*02 up to 100%, HLA-B*35 up to 100% and HLA-C*07 up to 50%) representing highly promising, broadly applicable antigen targets for immunotherapeutic approaches. Of note, 98% of these highly promising, broadly applicable antigen targets for immunotherapeutic approaches have never been described in previous large cohort CLL immunopeptidome studies[33,35].

In addition to the identification of novel off-the-shelf tumor-exclusive antigens, screening for naturally presented neoepitopes derived from tumor-specific mutations was performed for TOF$_{IMS}$ and Orbitrap HNSCC immunopeptidomes, using a sample-specific mutation database generated by next-generation whole exome sequencing of tumor and respective adjacent benign tissue. Of note, two naturally presented neoepitopes (LPADVTEDEF SFPQ_HUMAN$_{304-313}$ I308V and VYPLAFVLI MD13L_HUMAN$_{279-287}$ S282L) were identified in the TOF$_{IMS}$ dataset and validated using isotopically labelled synthetic peptides (Fig. 4g, h), whereas none of these neoepitopes were identified in the conventionally-acquired data. Together, TOF$_{IMS}$ MS provides a next generation immunopeptidomics method that facilitates the further prioritization of established TAAs and enables the identification of a vast array of previously undescribed non-mutated TAAs as well as the detection of naturally presented low abundant neoepitopes for cancer immunotherapy.

## Discussion

MS-based immunopeptidomics provides direct evidence of cellular processing and HLA-restricted presentation of peptide antigens, which is an indispensable prerequisite for their therapeutical use, in particular regarding the distorted correlation between gene expression and HLA-restricted antigen presentation[30,36–38]. In this study we implemented a next-generation MS-based immunopeptidome workflow using TOF$_{IMS}$ MS to expand benign immunopeptidomics reference databases and improve TAA discovery.

In line with improvements reported for other *omics* technologies, as well as with FAIMS technology for immunopeptidomics, TOF$_{IMS}$ methodology enabled high sensitivity and fast track peptide

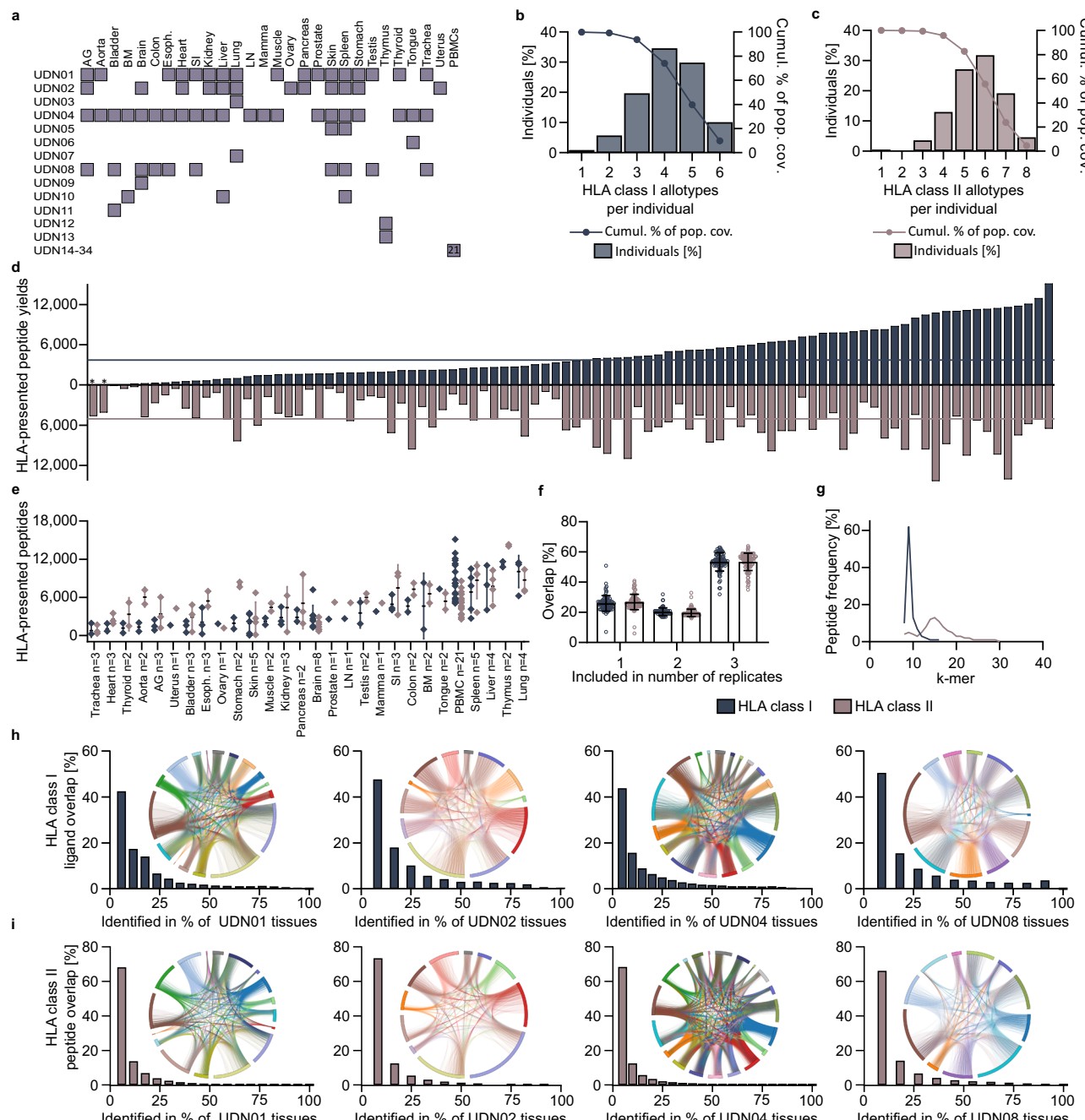

**Fig. 2 | Characterization of Benign_TOFIMS dataset. a** Sample overview included in the benign_TOFIMS dataset. **b, c** Population coverage of the HLA class I (**b**) and HLA class II (**c**) allotypes of the benign sample cohort compared to the world population determined by the IEDB population coverage tool[46]. Frequencies of individuals within the world population carry up to 6/8 allotypes (x-axis) are indicated as bars on the left y-axis. The cumulative percentage of population coverage is depicted as dots on the right y-axis. **d** HLA class I (top) and HLA class II (bottom) ligand yields of the primary benign tissue samples of various origins (*n* = 94). Median peptide yields are indicated as lines and samples marked with asterisk were only characterized for HLA class II-presented peptides. **e** Tissue specific HLA class I and HLA class II peptide yields of the benign_TOFIMS dataset (*n* = 94 biologically independent samples).

Mean peptide yields ± standard deviation are depicted with error bars. **f** Percentage of peptides included in one, two or three technical replicates within one sample in the benign dataset (*n* = 94 biologically independent samples). Mean peptide overlap ± standard deviation are depicted with error bars. **g** HLA class I and HLA class II peptide length distribution of the benign dataset. **h, i** HLA class I (**h**) and class II (**i**) peptide overlap of tissues originating from the same donor (*n* = 16 for UDN01, *n* = 12 for UDN02, *n* = 20 for UDN04 and *n* = 11 for UDN08) with corresponding chord plot analysis. AG Adrenal gland, BM Bone marrow, SI Small intestine, LN Lymph node, esoph. esophagus, UDN Universal donor number, HLA Human leukocyte antigen, cumul. Cumulative, pop. Population, cov Coverage. Source data are provided as a Source Data file.

identification. IMS provides a new dimension of separation with the CCS value as an additional peptide property, which has been suggested to improve statistical confidence in peptide identification[39,40]. TOF_IMS efficacy relies on releasing ions according to their ion mobility synchronized to the mass analysis via TOF, resulting in a high speed and

sensitive detection of HLA-presented peptides. The here established TOF_IMS workflow and detailed method development comprising an optimized IMS and mass range window allow for increased peptide identifications compared to previous reports[21,41]. The frequency of proteolytic peptide artefacts[29], originating during sample preparation,

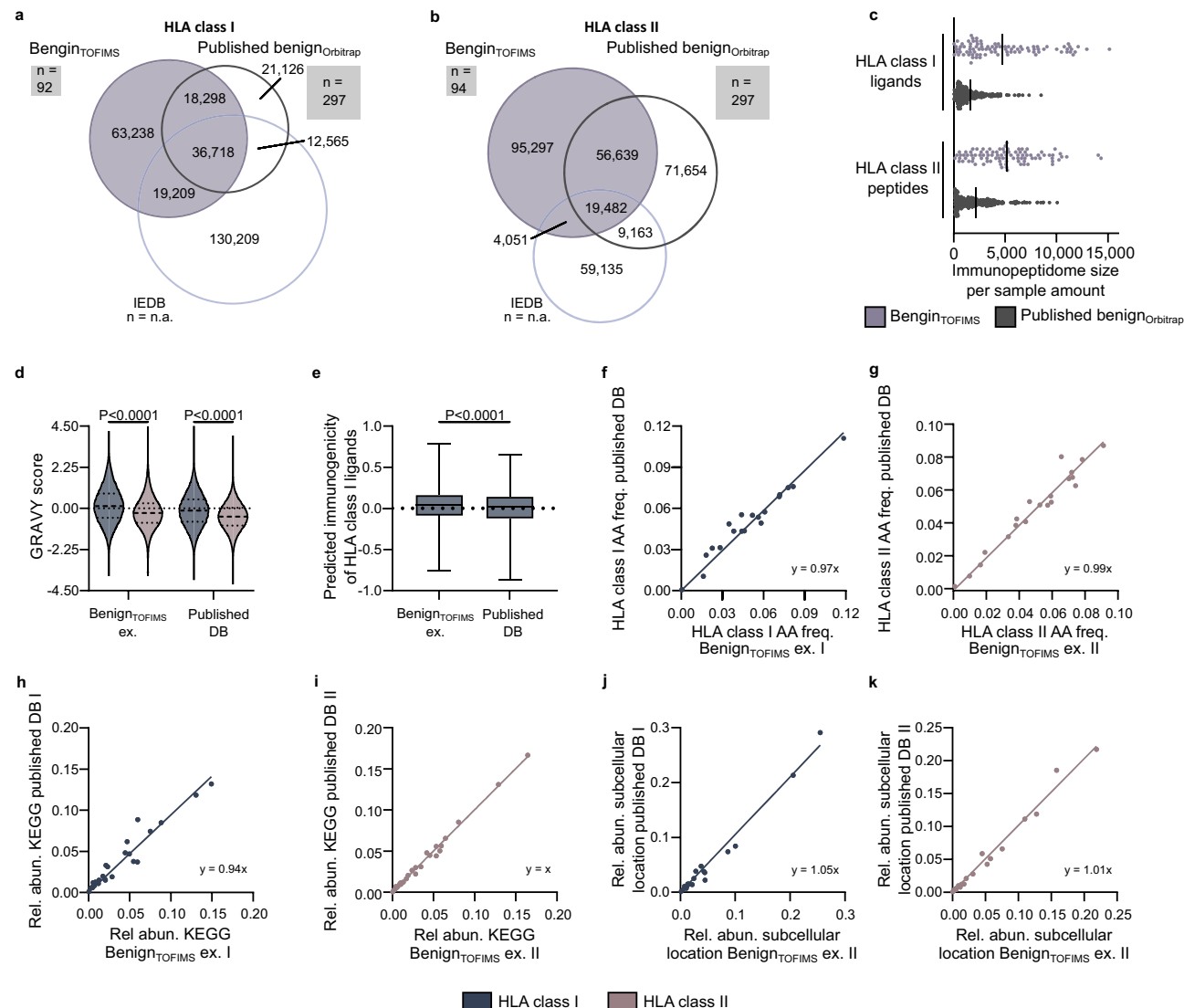

**Fig. 3 | Benign$_{TOFIMS}$ immunopeptidome dataset expands benign reference datasets of HLA-presented peptides. a**, **b** Overlap analysis of benign$_{TOFIMS}$ HLA class I (**a**) and HLA class II peptides (**b**) with public benign databases (IEDB[31] and Orbitrap-currated[8,30]). Square-size represent *n* included in respective datasets, correlating to sample size. **c** Number of HLA class I and HLA class II peptides identified per sample (*n* = 94) in benign$_{TOFIMS}$ or in published benign Orbitrap tissue datasets[8,30] (*n* = 297 samples). Mean indicated as line. **d** GRAVY score of benign$_{TOFIMS}$ exclusive HLA class I and HLA class II peptides compared to published benign databases[8,30,31]. Violin plots with median and 25th to 75th percentiles, unpaired Kruskal-Wallis t-test (*P* = 0). **e** Predicted immunogenicity of benign$_{TOFIMS}$ exclusive (*n* = 27,575) and published benign databases HLA class I ligands (*n* = 128,794). Immunogenicity was predicted using IEDB's online tool[64]. Box plots

show median immunogenicity with 25th to 75th percentiles and min/ max whiskers, unpaired, two-tailed Mann-Whitney test (*P* = 6 × 10$^{-96}$). **f**, **g** Amino acid composition of benign$_{TOFIMS}$ exclusive HLA class I (R$^2$ = 0.93, **f**) and HLA class II (R$^2$ = 0.99, **g**) peptides compared to published benign immunopeptidome databases. **h**–**k** Relative abundance of KEGG pathways (**h**, **i**) and subcellular locations (**j**, **k**) from benign$_{TOFIMS}$-exclusive HLA class I (R$^2$ = 0.93, **h** and R$^2$ = 0.98 **j**, respectively) and HLA class II (R$^2$ = 0.99, **i** and R$^2$ = 0.99 **k**, respectively) peptides compared to published benign databases. HLA Human leukocyte antigen, IEDB Immune Epitope Database, AA Amino acid, freq. frequencies, ex. Exclusive, DB Database, GRAVY Grand average of hydropathy, KEGG Kyoto Encyclopedia of Genes and Genomes, UDN Universal donor number, R$^2$ Goodness of fit. Source data are provided as a Source Data file.

was slightly increased in the TOF$_{IMS}$ immunopeptidome datasets, underscoring its ability to detect low abundant peptides. This makes TOF$_{IMS}$-based immunopeptidomics attractive for sample-limited applications, such as peptide identification from biopsies, micro-dissected tissues, and sorted cell populations[12,13].

Beyond the selection of immunogenic antigens from tumor tissue, tumor-exclusive presentation without representation of the respective antigen on benign tissue is of central importance to avoid on-target-off-tumor adverse events and enable tumor-directed immune targeting. In particular, for non-mutated TAA arising through differential gene expression and protein processing in malignant cells, and playing a central role as

immunotherapeutic targets in low-mutational burden tumor entities, knowledge of the immunopeptidome of benign tissue is a key prerequisite to define safe T cell-based cancer immunotherapies and has led to the development of benign immunopeptidome repositories[8,30,31,42–44]. However, despite the constant growth of these databases, the landscape of the whole benign immunopeptidome is still not covered and it is unclear when a saturation of peptide identifications will be reached.

Using TOF$_{IMS}$-based immunopeptidomics we build a novel benign tissue repository that substantially expanded these references, providing more than 150,000 previously undescribed HLA class I- and HLA class II-presented peptides from benign tissue origin. With this

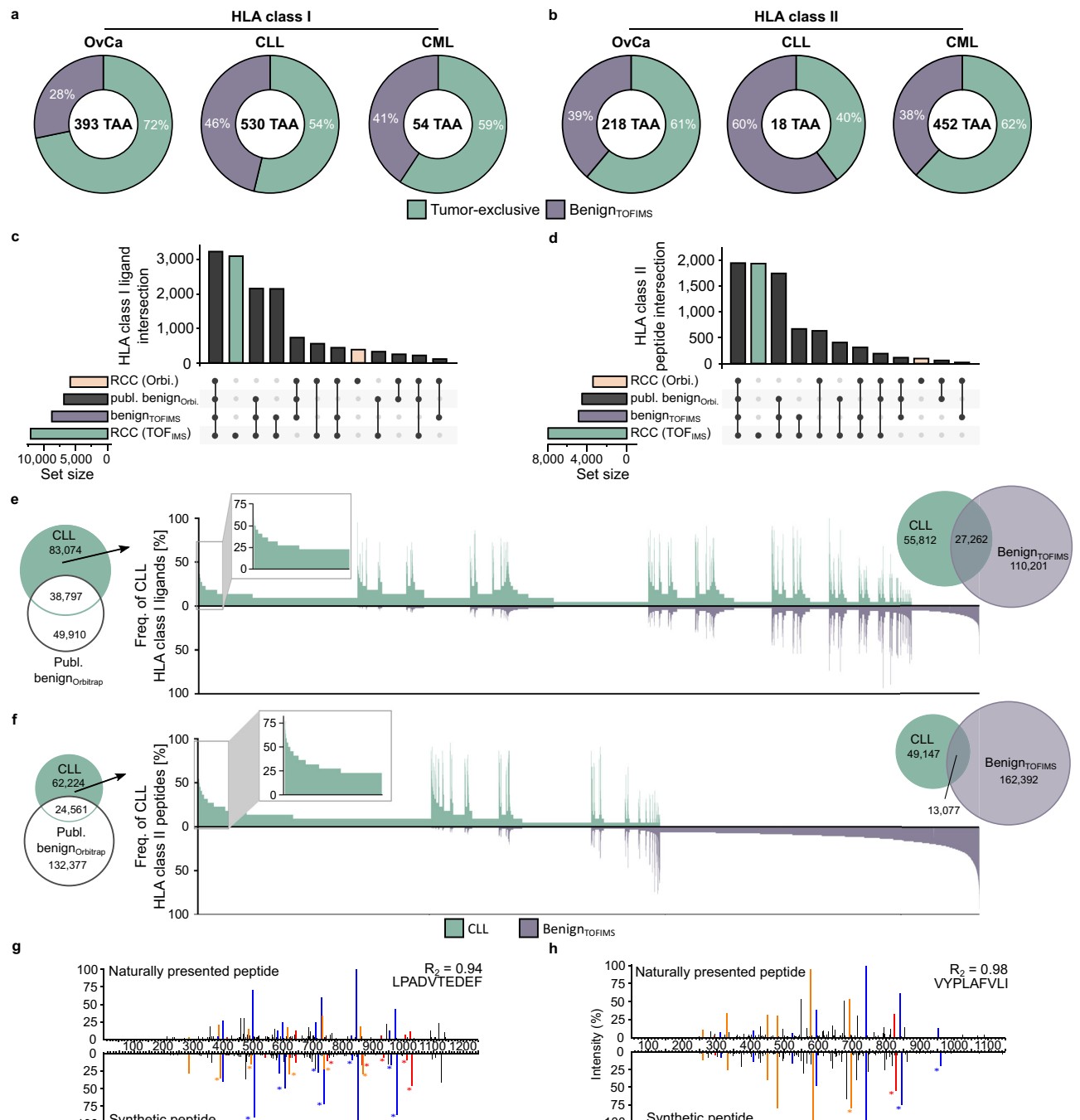

**Fig. 4 | Refined tumor-associated antigen identification using benign_TOFIMS MS.**
**a, b** Comparative immunopeptidome profiling of published HLA class I (**a**) and HLA
class II (**b**) tumor-associated antigens for OvCa[32], CLL[33] and CML[30] with benign_TOFIMS
tissue immunopeptidomes. **c, d** UpSet plots showing HLA class I ligand (**c**) and
HLA class II-presented peptide (**d**) intersection size between peptides identified from
RCC sample using TOF_IMS and Orbitrap with benign_TOFIMS and published benign
datasets. **e, f** Overlap analysis of HLA class I (**e**) and class II (**f**) peptides of primary
CLL samples (*n* = 22) with published benign datasets[8,30]. Comparative profiling of
HLA class I (**e**, right panel) and class II (**f**, right panel) peptides based on frequency
of CLL-presented peptides not included in published bening datasets with
benign_TOFIMS dataset. Frequencies of positive immunopeptidomes for the respec-
tive HLA ligands (x-axis) are indicated on the y-axis. For improved readability, HLA
ligands identified on < 5% of samples within the respective cohort were not

depicted in this plot. The magnified box represents the subset of CLL-associated
antigens showing CLL-exclusive and high-frequent presentation. **g, h** Mass
spectrometry-based neoantigen validation. The experimentally eluted mutation-
derived peptides LPADVTEDEF (SFPQ_HUMAN_304-313 I308V, (**g**) and VYPLAFVLI
(MD13L_HUMAN_279-287 S282L, (**h**) identified in HNSCC (above the x-axis) were
validated with the corresponding isotopically labelled synthetic peptide (mirrored
on the x-axis). Identified b-, y- and internal ions are marked in blue, red, and orange,
respectively. Ions containing the isotopically labelled amino acid are marked with
an asterisk. The indicated correlation coefficient ($R^2$) was determined with b and y
ions. TAA Tumor associated antigen, HLA Human leukocyte antigen, OvCa Ovarian
cancer, CLL Chronic lymphocytic leukemia, CML Chronic myeloid leukemia, RCC
Renal cell carcinoma, HNSCC Head and neck squamous cell carcinoma, publ.
published. Source data are provided as a Source Data file.

currently available reference databases were expanded by more than 50%, suggesting the benign$_{TOFIMS}$ database as future state-of-the-art benign reference for future selection and validation of non-mutated TAAs. Of note, HLA-presented peptides exclusively identified within the benign$_{TOFIMS}$ dataset showed significantly increased hydrophobicity, which was described as a hallmark of immunogenic T cell epitopes[45,46]. In line, predicted immunogenicity was higher in the benign$_{TOFIMS}$ exclusive peptides compared to published data[8,30,31], however both datasets present a general low median predicted immunogenicity which aligns with their benign origin. First application of these benign$_{TOFIMS}$ immunopeptidome dataset enabled the refinement of TAAs identified in previous studies for multiple tumor entities[30,32,33]. Moreover, TOF$_{IMS}$-based immunopeptidomics led to the identification of novel, high frequent non-mutated tumor-exclusive peptide antigens from primary CLL samples, highlighting the potential of this approach to expand the number of antigen targets to be applied in T cell-based immunotherapies.

In addition to non-mutated tumor antigens, neoepitopes arising from tumor-specific mutations have been identified in recent years as the main specificity of anti-cancer T cell responses induced by immune checkpoint inhibitors and were in turn suggested as prime candidates for T cell-based immunotherapy approaches[36,47,48]. In line, response to immune checkpoint inhibitors correlates with high mutational burden and neoepitope-based immunotherapies have been applied in various tumor patients[4,5,49]. However, the limited number and low abundance of somatic mutations that are ultimately translated, processed, and presented as HLA-restricted neoepitopes on the tumor cells[3,36,50–52] has hampered the MS-based identification and thus selection of optimal neoepitopes for cancer immunotherapy. Our TOF$_{IMS}$ immunopeptidomics workflow has led to the de novo identification of mutation-derived HLA-presented peptides, suggesting that the increased sensitivity of this approach might further improve the detection of naturally presented neoepitopes.

The limitations of this study comprise the lack of sample-specific sequencing, and the inability of the spectrum-annotation software to distinguish between isomers leucine and isoleucine and thus resulting possible inclusion of both sequences.

Together, our study provides a novel TOF$_{IMS}$-based immunopeptidome benign reference that enables the highly sensitive identification of HLA-presented peptides that will refine tumor antigen discovery.

## Methods
### Sample collection
Benign solid tissue samples were collected within 72 h post-mortem during routine autopsies at the University Hospital Zürich. Subjects included in this study were not diagnosed with any malignant disease. The tissue was annotated by board-certified pathologists, snap-frozen in liquid nitrogen and stored at −80 °C. Thymus samples were obtained from the University Children's Hospital Zürich and were removed during heart surgery for other medical reasons than cancer. Tumor samples and blood donation by-products for peripheral blood mononuclear cell (PBMCs) isolation from healthy individuals were collected at the University Hospital Tübingen. Informed consent was obtained in accordance with the Declaration of Helsinki protocol. The study was approved by and performed according to the guidelines of the local ethics committee (Req-2016-00604, EC-Nr. 2014-0699, PB_2017-00631, 424/2007B02, 373/2011B02, 431/2012BO2, 454/2016B02, 356/2017BO2, 406/2019B02). Donor characteristics are provided in Supplementary Data 2.

### Cell lines
EBV-transformed human B cell line JY (ECACC, England, UK 94022533; HLA-A*02, HLA-B*07, HLA-C*07) was cultivated in RPMI1640 with 10% heat-inactivated fetal bovine serum (FBS, Lonza, Basel, Switzerland)

and 1% penicillin/streptomycin (Merck, Darmstadt, Germany). Prior to sample preparation, cells were washed three times in phosphate-buffered saline (PBS) before 15 min centrifugation at 190 x g for harvest, and frozen at −80 °C at $1.2 \times 10^{7}$ (JY 1) and $8 \times 10^{6}$ cells (JY 2).

### Isolation of HLA ligands
HLA class I and HLA class II molecules were isolated by previously described immunoaffinity chromatography protocols[53] using the pan HLA class I-specific W6/32[54], pan HLA class II-specific Tü-39[55] and HLA-DR-specific L243[56] monoclonal antibodies. All antibodies were produced *in-house* at the Department of Immunology, University of Tübingen. For the ten samples included in the comparative immunopeptidome profiling between TOF$_{IMS}$ and Orbitrap MS acquired data, 50% of each sample was analyzed per device in technical triplicates (two technical replicates for the HNSCC sample) with 5 µL per injection, respectively. For the immunopeptidome analysis of the TOF$_{IMS}$ benign dataset ($n = 94$), 60% of the sample were injected in three technical replicates with 5 µL per injection.

### TOF$_{IMS}$ mass spectrometric data acquisition
For the method development various parameters were tested as indicated in Supplementary Data 1 using the Bruker Daltonic's timsTOF Pro device, a MS approach combining the technologies trapped ion mobility spectrometry (TIMS) and parallel accumulation-serial fragmentation (PASEF) coupled to a time-of-flight (TOF) mass spectrometer[15,16,57]. Samples included in the TOF$_{IMS}$-Orbitrap MS comparison as well as the TOF$_{IMS}$ benign dataset were analyzed using the method described in Table 1. Peptide separation was performed on Bruker Daltonic's nanoElute LC system using an acclaim TM PepMap (Thermo Fisher Scientific, Waltham, USA) and a 75 µm x 25 cm Aurora Series emitter column (IonOpticks, Fitzroy, Australia). Peptides were separated along a gradient ranging from 0% to 95% Solvent B (AcN with 0.01% FA) over the course of 60 min with consecutive ramps from 0% to 32% (30 min) and 32% to 40% (15 min), followed by two 5 min ramps to 60% and 95%, respectively. Eluting peptides were subsequently analyzed in the on-line coupled trapped ion mobility spectrometry and time-of-flight mass spectrometer timsTOF Pro (Bruker Daltonics, Billerica, USA) equipped with a CaptiveSpray ion source using a data-dependent acquisition mode (DDA).

### Orbitrap mass spectrometric data acquisition
The orbitrap-based MS analysis was performed as described previously in ref. 58. Peptides were separated by nanoflow high-performance liquid chromatography using a Thermo Fisher Scientific's Ultimate 3000 RSLC Nano UHPLC system, loaded with 1% AcN 0.05% TFA on a 75 µm x 2 cm Acclaim PepMap 100 C18 Nanotrap column at a flow rate of 4 mL/min for 10 min following a separation step using a 50 µm x 25 cm PepMap RSLC C18 column with a particle size of 2 µm. Peptides were eluted at a gradient ranging from 2.4% to 32% AcN over 90 min. Eluted peptides were analyzed in the on-line coupled Orbitrap Fusion Lumos mass spectrometer (Thermo Fisher Scientific, Waltham, USA) equipped with a nano electrospray ion source in DDA acquisition mode employing a top-speed collisional-induced dissociation (CID, HLA class I-presented peptides, normalized collision energy 35%) or higher-energy collisional dissociation (HCD, HLA class II-presented peptides, normalized collision energy 30%) fragmentation method. Mass range for HLA class I-presented peptide analysis was set to 400−650 m/z with charge states 2+ and 3+ selected for fragmentation. For HLA class II-presented peptide analysis mass range was limited to 400−1000 m/z with charge states 2+ to 5+ selected for fragmentation.

### Whole exome sequencing
Whole exome sequencing was performed from the same snap-frozen tumor tissue used for the immunopeptidomics analysis via Illumina NovaSeq 6000 by an external provider (CeGaT GmbH) with a target

read length of 100 bp. Single-nucleotide polymorphism mutations were excluded by comparing with adjacent benign tissue from the same patient identifying 271 tumor-exclusive mutations.

## Database search

Data processing was performed using PEAKS Studio 10.6 (Bioinformatic Solutions Inc.). All samples were searched against a database containing 20,385 reviewed human UniProt entries downloaded on 14.10.2020. For the HNSCC the corresponding mutations were added to the reference database. The enzyme specificity was set to none, precursor peptide mass error tolerances were set to 5 ppm (orbitrap MS data) or 20 ppm (TOF$_{IMS}$ data) and 0.02 Da for fragment ions. Oxidized methionine was set as variable modification, with three possible modifications allowed per peptide. Peptide lengths were set to 8–16 amino acids for HLA class I and 8–30 amino acids for HLA class II. A 1% false discovery rate (FDR) was calculated using a decoy database search approach. HLA class I identified peptides were further annotated as ligands using SYFPEITHI 1.0[59] and netMHCpan4.1[60] using the sample's respective HLA allotypes.

## Synthesis of isotope-labeled peptides

Isotopically labelled peptides were synthesized using the standard 9-fluorenylmethyl-oxycarbonyl/tert-butyl strategy in a Liberty Blue Automated Peptide Synthesizer (CEM, Kamp-Lintfort, Germany). Peptides were cleaved from the resin using a TFA/triisopropylsilane/water (95%/2.5%/2.5% by vol.) mixture for 1 h, after which peptides were precipitated with diethyl ether and washed with diethyl ether thrice before resuspension in water and lyophilization. Identity and purity were determined via C18-HPLC and LTQ Orbitrap XL MS (both Thermo Fisher Scientific).

## Spectrum validation

Spectrum validation of the experimentally eluted peptides was performed by computing the similarity of the spectra with corresponding isotopically labelled synthetic peptides measured in a complex matrix. The spectral correlation coefficient was calculated between the b and y ions of the MS/MS spectra of the eluted and synthetic peptide[61].

## Identifying proteolytic fragments

To identify proteolytic peptide artifacts, a statistical method[29] that calculates the proposed protein coverage ratio, peptide coverage ratio, and HLA ligand propensity scores for each peptide, was implemented. An expectation-maximization algorithm was used to deconvolute the Gaussian mixture into two distributions, defining a threshold for each of the scores with 0.05 FDR. A peptide was classified as proteolytic if it superseded two of the three thresholds, similar to the original publication[29]. The three parameters were computed based on the ten comparative samples. The statistical method implementation can be found at [https://github.com/AG-Walz/proteolytic_degradation_timstof_orbitrap].

## Comparison of benign$_{TOFIMS}$ immunopeptidome data with published datasets

The timsTOF Pro benign dataset was compared with published benign orbitrap databases comprising immunopeptidomes derived from benign tissues (HLA Ligand Atlas)[8] and hematological cells[30] as well as the IEDB database[31], respectively. The HLA Ligand Atlas peptides were filtered to medium and strong binders, for the IEDB healthy, linear peptides of human source obtained through MHC ligand assay were included. Published TAAs for OvCa[32], CLL[33] and CML[30] were retrieved from previous publications.

## Statistical analysis

Overlap and UpSet plot analysis were performed using BioVenn[62] and UpSetR Shiny[63]. Grand average of hydropathy (GRAVY score) was calculated using the GRAVY calculator (https://www.gravy-calculator.de). For the ranked reported area analysis, peptides with a reported area were normalized according to sample and device specificity. Frequency-based overlap analysis between CLL identified peptides and the benign$_{TOFIMS}$ dataset show all identified peptides with a frequency > 5% and < 5% if founs on both the CLL and benign$_{TOFIMS}$ dataset. The population coverage and immunogenicity (only 9-mers) were predicted using IEDB's population and immunogenicity prediction tools[46,64] [http://tools.iedb.org/population/ and http://tools.iedb.org/immunogenicity/]. The subcellular location analysis was annotated according to the human protein atlas[65] [https://proteinatlas.org/about/downloads] version 22.0, Ensembl version 103.38. Kyoto Encyclopedia of Genes and Genomes (KEGG)[66] annotation was performed according to the 106.0 release. Chord plots were performed using pycirclize version 0.4.0 [https://github.com/moshi4/pyCirclize] with Shimoyama et al. library [https://github.com/moshi54/pyCirclize]. All figures and statistical analysis were generated using GraphPad Prism 9.2.0 (GraphPad Software) or MS Office Excel 2019. Data are displayed as mean ± SD, box plots as median with 25th or 75th percentiles and min/max whiskers. Continuous data were tested for distribution and individual groups were tested by use of two-sided Fisher's exact test, unpaired t-test, unpaired Mann-Whitney-U-test, Kruskal-Wallis test, or paired Wilcoxon signed rank test, all performed as two-sided tests. If applicable, adjustment for multiple testing was made. $P$-values of < 0.05 were considered statistically significant.

## Data availability

The mass spectrometry data have been deposited in the ProteomeXchange Consortium database [https://www.proteomexchange.org/] via the PRIDE partner repository[67] under dataset identifier PXD03878. Source data are provided with this paper. The data used for comparative purposes can be found at HLA ligand atlas [https://hla-ligand-atlas.org/welcome], IEDB [https://www.iedb.org/] (selecting epitope linear peptide, epitope source human, assay MHC ligand, MHC restriction, disease healthy) and the data set provided with Bilich et al. (2019 in Blood). Source data are provided with this paper.

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

## Acknowledgements

We thank R. Agrusa, C. Falkenburger, and U. Wulle for excellent technical support. This work was supported by the Deutsche Forschungsgemeinschaft under Germany's Excellence Strategy (Grant EXC2180-390900677 (H.-G.R., H.R.S. and J.S.W.)), the German Cancer Consortium (DKTK) (H.-G.R., H.R.S. and J.S.W.), the Deutsche Forschungsgemeinschaft (DFG, German Research Foundation, Grant WA4608/1-2 (J.S.W.), the Wilhelm Sander Stiftung (Grant 2016.177.3 (J.S.W.)), the José Carreras Leukämie-Stiftung (Grant DJCLS 05 R/2017 (J.S.W.)), the Else Kröner Fresenius Stiftung, Translatorik Programm (Grant 2022_EKTP03 (J.S.W.)), the Deutsche Krebshilfe (German Cancer Aid, 70114948 (J.S.W.)), the Zentren für Personalisierte Medizin (ZPM, J.S.W.), the Applied Clinical Research program (AKF, (J.S.W.)), and the Fortüne Program of the University of Tübingen (Fortüne number 2451-0-0 (S.M.S.)). We acknowledge support from the Open Access Publication Fund of the University of Tübingen.

## Author contributions

N.H.G., A.N., J.B., L.M. and J.S.W. conceptualized this study. N.H.G., L.M., S.M.S, A.D. and M.W. performed immunopeptidome experiments. N.H.G., S.L., J.S. and M.L.D. performed bioinformatic analysis. S.M.S., M.C.N., P.-S.M., H.L., M.H.-H., R.M., J.H., A.S., M.H.-H., R.M., J.S.H. and H.R.S. collected the samples included in this study. R.K. performed HLA typing analysis. A.N., J.B., H.-G.R. and J.S.W. supervised this study. All authors contributed to the article and approved the submitted version.

## Funding

## Competing interests

The authors declare no competing interests.

## Additional information

[1]Department of Peptide-based Immunotherapy, University and University Hospital Tübingen, Tübingen, Germany. [2]Institute for Cell Biology, Department of Immunology, University of Tübingen, Tübingen, Germany. [3]Cluster of Excellence iFIT (EXC2180) "Image-Guided and Functionally Instructed Tumor Therapies", University of Tübingen, Tübingen, Germany. [4]German Cancer Consortium (DKTK) and German Cancer Research Center (DKFZ), partner site Tübingen, Tübingen, Germany. [5]Department of Otorhinolaryngology, Head and Neck Surgery, University of Tübingen, Tübingen, Germany. [6]Neuroscience Center Zürich (ZNZ), University of Zürich and ETH Zürich, Zürich, Switzerland. [7]Clinical Neuroscience Center and Department of Neurosurgery, University Hospital and University of Zurich, Zürich, Switzerland. [8]Department of Neurosurgery, Cantonal Hospital St. Gallen, Zürich, Switzerland. [9]Quantitative Biology Center

(QBIC),  University of Tübingen, Tübingen, Germany. [10]Department of Hematology, Oncology, Clinical Immunology and Rheumatology, University Hospital Tübingen, Tübingen, Germany. [11]Pediatric Stem Cell Transplantation, University Children's Hospital Zürich, Zürich, Switzerland. [12]Neuroimmunology and MS Research, Neurology Clinic, University and University Hospital Zürich, Zürich, Switzerland. [13]Department of Urology, University Hospital Tübingen, Tübingen, Germany. [14]Clinical Collaboration Unit Translational Immunology, German Cancer Consortium (DKTK), Department of Internal Medicine, University Hospital Tübingen, Tübingen, Germany. [15]Present address: Clinical Collaboration Unit Translational Immunology, German Cancer Consortium (DKTK), Department of Internal Medicine, University Hospital Tübingen, Tübingen, Germany. ✉e-mail: juliane.walz@med.uni-tuebingen.de

