## [Peer Review File · Nature Communications]

REVIEWER COMMENTS

Reviewer #1 (Remarks to the Author):

The authors have performed Immunopeptidome analysis using two different mass spectrometry systems and claim that the Bruker timsTOF system has better performance for this analysis. Using the timsTOF system, they have successfully identified more than 150,000 antigenic peptides in 94 tissue samples.

Although the amount of information obtained is enormous, they did not develop any new analytical techniques. Therefore, it looks to be a report of manufacturer's performance evaluation test using an existing mass spectrometer. Indeed, There is no consideration or verification experiment that would generate a new immunological or biological concept from the obtained data, and it must be said that there is little novelty in this study.

More critically, a fair comparison of the key technical aspects was NOT performed. The timsTOF system recommended by the authors uses ion mobility fractionation, however, the Orbitrap system used in the study uses a model that does not have ion mobility.

This is an inappropriate comparison, and it would be unfair if the Orbitrap Fusion Lumos-FAIMS Pro were not used for comparison. In this regard, I believe that this study should be reconstructed as a scientific paper without misleading impression.

Reviewer #2 (Remarks to the Author):

The manuscript "timsTOF mass spectrometry-based immunopeptidomics refines tumor antigen identification" by Hoenisch Gravel et al. evaluates and optimises the use of an MS platform by Bruker Daltonics for immunopeptidomics workflows. It further compares this system with a so far broadly applied technology platform from Thermo Scientific. Both instruments used here are not the state-of-the-art machines from either company, and neither of the technology has been co-developed by the authors.

Whilst I believe this is a very carefully pursued optimization study, with excellent comparative experiments performed for the timsTOF system, I feel very uncomfortable about the direct comparison between the LC-MS systems of two different vendors (Bruker Daltonics and Thermo scientific), and do not think that the presented comparison is beneficial for the research community. The two different platforms use different chromatography, and different mass analysers. The Thermo Scientific platform used here lacks an ion mobility device. However, for the latter system such a device is available, and it would have been a minimal requirement to include it in such a comparative evaluation.

It is unclear why the extended benign datasets are of value for cancer antigen discovery (which they clearly are), as authors highlight identification of neoantigens, which have are the result of tumour-specific mutation and therefore do not benefit from the additional analysis of benign tissues.

Therefore, I believe this manuscript is not currently suitable for publication in Nature Communications. I would suggest a removal of comparative data, and the publication of the highly relevant benign tissue data as an extension of the HLA ligand atlas first published by the authors in 2021.

Reviewer #3 (Remarks to the Author):

T cell recognition of human leukocyte antigen (HLA) presented tumor associate peptides is central for cancer immune surveillance. This manuscript describes the use of timsTOF with ion mobility separation-based MS for immunopeptidomics profiling. Comparisons between MS data generated via IMS timsTOF versus Orbitrap were also performed. The authors

found a significant gain of immunopeptide acquisitions through the timsTOF. The authors then analyzed MHC I/II peptidomes from up to 94 benign tissues and 4 primary malignant samples and identified 150,000 new peptides.

1. The authors should describe in more details how the timsTOF specifically allow for the gain of identifications compared to the Orbitraps. For example, are there any physiochemical properties unique to the immunopeptides that render them more challenging to be detected without IMS capability? How are detections hampered by Orbitrap alone. At present it is hard to figure out what to conclude from the comparison with the Orbitrap. It is clear different instruments have different strengths and the timsTOF is a faster machine. Some of the performance gains from the timsTOF data set might also have been reached using better separation and longer acquisition time on the Orbitrap.

2. The technical or methodological novelty of the study is not clearly stated. Even though it is a relatively newer technology than the Orbitraps, the Bruker timsTOF Pro has been introduced since 2017, and it is widely utilized by the proteomics community nowadays, and so is not exactly a novel MS instrument in 2023. MHC-I immunopeptide data using a timsTOF instrument with PASEF have been demonstrated before (Feola et al. eLife 2022). The authors should cite these studies and better describe how the current approach differs.

3. The manuscript will benefit from additional analyses focusing more on the new peptides identified and their biological relevance instead of being heavily centered on the numbers of identification. Currently I feel the manuscript mostly describes the availability and depth of a data set without a lot of analysis on what it all means.

4. Are there any quantitative information that can be drawn from the MS data? Since the authors mentioned low abundance peptides at several places, some relative quantification could potentially offer some insights into the relative abundance of these peptides within individual and across the different samples. Are there significant variability among the different biological replicates? How does the reproducibility compare between the Orbitrap and timsTOF in terms of signal intensity across samples and also identification?

5. The lack of personalized genome specific protein sequence database (with SAAV or tumor mutations) for most of the samples (except the HNSCC samples) is in my view a technical weakness that limits the utility of the data set. It perhaps also raises some question about identification fidelity when some mutant peptides are probably present that cannot be identified with the database used. For the HNSCC mutations it was not clearly explained whether the exome data were from matching samples as the immunopeptidome data.

6. The results section stated that the current data set rejected up to 40% of previously published OvCa TAAs. The authors may want to expand on what kind of peptides were rejected, or whether the retained peptides now constitute true TAA. It is already recognized that neoepitopes provide better TAAs for immunotherapy (one might assume the current data set demonstrates rather the danger of using non-mutated peptides as TSA/TAA instead since if sensitivity is unlimited then presumably every protein/peptide might be expected to be loaded on an MHC).

7. Since MHC peptides are usually considered challenging to identify, the authors may want to compare different search engines and post-processing workflows, and also consider whether 1% FDR cutoff as opposed to spectrum- or peptide- specific error probability is sufficiently stringent.

Reviewer #4 (Remarks to the Author):

In their manuscript entitled "timsTOF mass spectrometry-based immunopeptidomics refines tumor antigen identification", Gravel et al. compare the performance of a timsTOF mass spectrometer-based workflow to an Orbitrap-based workflow. The authors apply the developed TimsTOF method towards the characterization of the MHC class I and MHC class

II ligandomes of 94 benign tissues and show principal applicability of the workflow towards neoantigen discovery.

The manuscript is well written and figures are of high quality.

Presentation of the data is clear, but some results are shown at a very highly aggregated level that is in some cases potentially prone to misinterpretation.

Overall, the work is of high significance to the field, but some analyses need to be explained in more detail and supported by supplementary figures and tables to enable them to be easily reproduced by other researchers.

I have several major points that the authors should address in a revised version:

1. Claims of novelty should be toned down, as the application of timsTOF to immunopeptidomics is not completely novel (DOI: 10.3389/fimmu.2022.932252), Casasola-LaMacchia et al. described the identification of 12.000 HLA-II associated peptides from dendritic cells.

2. Figure 1d: This is described as the distributions of peptide m/z from a JY-2 sample, but the total number of peptides in this panel seems to be 10x higher than in the other panels.

2. The authors compare the timsTOF instrument to an Orbitrap Fusion Lumos platform. Klaeger et al. previously showed a beneficial effect of FAIMS ion mobility for Orbitrap instruments (DOI: 10.1016/j.mcpro.2021.100133), which significantly enhanced sensitivity. To make a fair comparison, the authors ideally should compare timsTOF vs. a FAIMS-equipped Orbitrap platform on selected samples.

3. Figure 2a: It is not clear how these numbers (ligands per tissue) were derived.

Are these identifications pooled from different patients?

What is the number of ligands found in each patient sample?

How reproducible are the results, i.e. what is the overlap between technical replicates in the tissue samples?

The overall ligand numbers across 94 tissues reported are quite impressive, but the authors also need to show what would be expected from the analysis of individual tissues/samples, ideally on technical replicate level, as a supplementary figure and provide a supplementary table containing the numbers of peptides identified in each of the samples.

4. Figure 3c: The authors seem to compare the results here on a very global level - it is not at all clear from which sample(s) the data are derived. Analyzing the data on Proteomexchange indicates that some timsTOF datasets have fewer than 500 identifications.

5. What is the overlap in presented ligands between different tissues from the same patient? This analysis should be easy to derive from the data and could provide important insights regarding tissue-specificity of antigen processing.

Minor comments:

1. the specific coordinates of isolation polygons (V2/V3/V4) used need to be defined to enable other groups to reproduce the findings.

2. Line 176-182 this statement should be modified, as it reads somewhat like a Bruker marketing brochure.

3. according to the search results deposited on ProteomeXchange, some peptides were not mapped to a source protein, or the respective identification is not provided. Were these peptides counted as identified? Please explain.

4. Some spectra seem to be double annotated:

e.g. in file

210831_NHG_benign_UDN04_Bladder_Tue39L243_20%_3RepsCoProcessed_peptide
several precursor IDs are annotated with two different peptide sequences that differ only
by a Leucine/Isoleucine replacement. This seems to artificially inflate the numbers
reported.

5. In the PEAKS reports, a significant proportion of peptides (20-30%) has a reported area
of 0 in all replicates. Were these counted as identified? Please clarify / explain.

6. How do the authors treat peptides that do not match the expected length distribution of
MHC ligands? Do the authors e.g. count 9-11 mers as identified ligands for MHC class II or
15+ mers as ligands for MHC I? Please clarify and discuss and provide the decision criteria.

7. The cycle time for the 8 PASEF method indicated in Supplementary Table 2 seems to be
incorrect.

Reviewer #1 (Remarks to the Author):

Comment #1: *The authors have performed Immunopeptidome analysis using two different mass spectrometry systems and claim that the Bruker timsTOF system has better performance for this analysis. Using the timsTOF system, they have successfully identified more than 150,000 antigenic peptides in 94 tissue samples.*

Author's reply: We thank the reviewer for the kind review and highly appreciate the input on how to improve our manuscript. Please find below a detailed point-to-point reply to the specific questions and issues raised.

Comment #2: *Although the amount of information obtained is enormous, they did not develop any new analytical techniques. Therefore, it looks to be a report of manufacturer's performance evaluation test using an existing mass spectrometer. Indeed, there is no consideration or verification experiment that would generate a new immunological or biological concept from the obtained data, and it must be said that there is little novelty in this study.*

Author's reply: We apologize for presenting not clearly enough the scope, intension and thus relevance and novelty of our work and further would like to emphasis that it was never our intention to conduct a competitive comparison of devices from different vendors.

The field of immunopeptidomics is of central importance to elucidate the antigenic landscape of HLA-presented peptides, which in recent years have gained increasing relevance in the development of immunotherapies for the treatment of various disorders, comprising cancer, autoimmune and infectious diseases (Chong *et al.* Nat. Biotechnol. 2022). Thus, the intention of this manuscript was to show how the advance of more sensitive mass spectrometric (MS) technology can be translated to the field of immunopeptidomics. MS technologies and methods are mostly developed, validated, and tested for proteomics, commonly analyzing tryptic peptides. This also holds true for the use of ion mobility spectrometry (IMS) in MS, where immense advances have been described in the field of proteomics so far but only very limited data exists for the use in immunopeptidomics. HLA-presented peptides show, compared to tryptic peptides, a less heterogenous length distribution and less defined characteristics, making their MS identification considerably challenging and accordingly require specifically tailored MS methods.

Beside the development and characterization of a timsTOF method (referred to as TOF_{IMS} in the revised manuscript) for the application in immunopeptidomics, this work provides an enormous pool of newly characterized peptides from benign tissue origin as (bening_{TOFIMS}) a comparative resource for future studies in the field and shows their first application for the definition of tumor associated antigens.

Within the revised manuscript, we stated more clearly the scope of this manuscript as well as the relevance of the results (please refer to line 44 - 50). As described in detail below (reply to comment #3), we removed any insinuations of reports of manufacturer's performance evaluation testing as well as any claims of superiority of a specific manufacturer's device.

Comment #3: *More critically, a fair comparison of the key technical aspects was NOT performed. The timsTOF system recommended by the authors uses ion mobility fractionation, however, the Orbitrap system used in the study uses a model that does not have ion mobility. This is an inappropriate comparison, and it would be unfair if the Orbitrap Fusion Lumos-FAIMS Pro were not used for comparison. In this regard, I believe that this study should be reconstructed as a scientific paper without misleading impression.*

Author's reply: We thank the reviewer for making this point. We completely understand and agree with the concerns raised by all reviewers regarding the direct comparison between timsTOF (referred to as TOF_{IMS} and Orbitrap without ion mobility. We would like to emphasize that it was never our intention to conduct a competitive comparison of devices from different vendors. Indeed, our intention was to provide immunopeptidome analysis of paired samples measured with the newly introduced TOF_{IMS} based method with the standard technology currently used in the field of immunopeptidomics. When performing a PubMed search (status April 2023) for the MS technologies/ instruments used in in the last five years (2018 - 2023) for immunopeptidomics/ ligandomics, >80% were conducted using an Orbitrap device (see Reviewer only Figure 1), including also published benign immunopeptidomic databases (e.g. IEDB).

Reviewer only Figure 1 | MS devices used in the field of immunopeptidomics. Percentage of publications on PubMed using different MS devices from 2018-2023 (April). The absolute number of publications is indicated above the bar.

In response to these concerns, we have completely restructured our manuscript in order to present more clearly the focus of the manuscript, which is (i) the introduction of an immunopeptidomic method for TOF_{IMS}, (ii) the development of a novel benign_{TOFIMS} immunopeptidome dataset, and (iii) its application in antigen target definition. The immunopeptidome analysis of paired samples on timsTOF and Orbitrap, which was a substantial part of the first manuscript version, was now integrated in the method development

section, has been significantly reduced and toned down with avoidance of any comparative statements and claims of superiority. In addition, we removed any claims of superiority from the revised introduction and discussion section and included a detailed discussion on the benefits of IMS use in general in the discussion section of the manuscript (please refer to line 190 - 199). Moreover, we have revised the naming of both devices to TOF_{IMS} and FT-ICR to focus more on the technical features instead of promoting a specific brand or instrument.

Our revised manuscript now emphasizes the methodological development and the biological relevance of its' usage in immunopeptidomics. To this end we have included multiple new analyses and experiments that are presented in the reconstructed Figures 3, 4, 5 and Supplementary Figures 3 and 4.

We would like to thank the reviewer again for the input on how to further improve our manuscript and hope that you are content with the additional experiments, data analyses and revisions we made to the manuscript.

Reviewer #2 (Remarks to the Author):

Comment #1: The manuscript “timsTOF mass spectrometry-based immunopeptidomics refines tumor antigen identification” by Hoenisch Gravel *et al.* evaluates and optimises the use of an MS platform by Bruker Daltonics for immunopeptidomics workflows. It further compares this system with a so far broadly applied technology platform from Thermo Scientific. Both instruments used here are not the state-of-the-art machines from either company, and neither of the technology has been co-developed by the authors.

Author’s reply: We thank the reviewer for the appreciated input and feedback to improve our manuscript. Whilst we agree with the reviewer that the instruments used in this study do not represent the latest devices on the broader mass spectrometer (MS) market (e.g. for proteomics), the orbitrap platform still is mostly used in the specific field of immunopeptidomics, in particular the Orbitrap Lumos and Orbitrap QE without ion mobility capacity. This becomes even more evident in a PubMed analysis (keyword search immunopeptidomics and ligandomics, only articles containing primary MS data were considered, April 2023) of the commonly used LC-MS systems in immunopeptidome research that found >70 % (n = 174) of immunopeptidome analyses in the past five years were conducted using either orbitrap system (~1 % with high-field asymmetric waveform ion mobility spectrometry (FAIMS), please refer to Reviewer only Figure 1).

Despite being introduced in 2017, the Bruker timsTOF system (referred to as TOF_{IMS} in the revised manuscript) was only used in ~1 % (n = 3, first in late 2022) of publications and thus still is novel to the field of immunopeptidomics. As seen with the FAIMS technology (Klaeger *et al.* Mol Cell Proteom 2022 and Minegishi *et al.* 2021 Commun Biol. 2022) and within this manuscript, immunopeptidomics will benefit from the use of IMS coupled MS technologies.

Reviewer only Figure 1 | MS devices used in the field of immunopeptidomics. Percentage of publications on PubMed using different MS devices from 2018-2023 (April). The absolute number of publications is indicated above the bar.

As the field of immunopeptidomics is of central importance in understanding the antigenic landscape of HLA-presented peptides, which in recent years have gained increasing relevance in the development of immunotherapies for the treatment of various disorders, comprising cancer, autoimmune and infectious diseases (Chong *et al.* Nat. Biotechnol. 2022), the intention

of this manuscript was to show how the advance of more sensitive MS technology can be translated to the field of immunopeptidomics.

As mentioned above MS methods and technologies are mostly developed, validated, and tested for proteomics, commonly analyzing tryptic peptides. This also holds true for the use of ion mobility spectrometry (IMS) in MS, where immense advances have been described in the field of proteomics so far, but only very limited data exists for the use in immunopeptidomics. HLA-presented peptides show, compared to tryptic peptides, a less heterogenous length distribution and less defined characteristics, making their MS identification considerably challenging and accordingly requires specifically tailored MS methods. To further delineate the eligibility of TOF_{IMS} for HLA-presented peptide analysis, in particular to provide novel insights to and expansion of the immunopeptidome landscape of large-scale benign and malignant datasets, we performed the alignment of paired primary benign and malignant samples (n = 10, Supplementary Data 2) analyzed using TOF_{IMS} and the current state-of-the-field (FT-ICR) technology, applied in current immunopeptidome references, which is as outlined above the orbitrap technology (Marcu *et al.* J Immunother Cancer 2021, Kraemer *et al.* Front Immunol 2022, Nicholas *et al.* Immunology 2023).

Beside the development and characterization of TOF_{IMS} for the application in immunopeptidomics, this work provides an enormous pool of newly characterized peptides from benign tissue origin as a comparative resource for future studies in the field and show their first application for the definition of tumor-associated antigens.

Within the revised manuscript, we stated more clearly the scope of this manuscript as well as the relevance of the results (please refer to line 45 - 51). As described in detail below (reply to comment #2 and #3), we removed any insinuations of reports of manufacturer's performance evaluation testing, as well as any claims of superiority of a specific manufacturer's device.

Comment #2: *Whilst I believe this is a very carefully pursued optimization study, with excellent comparative experiments performed for the timsTOF system, I feel very uncomfortable about the direct comparison between the LC-MS systems of two different vendors (Bruker Daltonics and Thermo scientific), and do not think that the presented comparison is beneficial for the research community.*

Author's reply: We thank the reviewer for making this point. We completely understand and agree with the concerns raised by all reviewers regarding the direct comparison between timsTOF and Orbitrap without ion mobility. We would like to emphasize that it was never our intention to conduct a competitive comparison of devices from different vendors. Indeed, our intention was to provide immunopeptidome analysis of paired samples measured with the

newly introduced TOF_{IMS} based method with the standard technology currently used in the field of immunopeptidomics (please also refer to author's reply to comment#1).

In response to these concerns we have completely restructured our manuscript in order to present more clearly the focus of our manuscript, which is (i) the introduction of an immunopeptidomic methods for timsTOF, (ii) the development of a novel benign_{TOFIMS} immunopeptidome dataset, and (iii) its application in antigen target definition. The immunopeptidome analysis of paired samples on timsTOF and Orbitrap, which was a substantial part of the first manuscript version, was now integrated in the method development section, has been significantly reduced and toned down with avoidance of any comparative statements and claims of superiority. In addition, we removed any claims of superiority from the revised introduction and discussion sections and included a detailed discussion on the benefits of IMS use in general in the discussion section of the manuscript. (please refer to abstract, results and discussion sections). Moreover, we have revised the naming of both devices to TOF_{IMS} and FT-ICR to focus more on the technical features instead of promoting a specific brand or instrument.

Our revised manuscript now emphasizes the methodological development and the biological relevance of its' usage in immunopeptidomics. To this end we have included multiple new analyses and experiments that are presented in the reconstructed Figures 3, 4, 5 and Supplementary Figures 3 and 4.

Comment #3: *The two different platforms use different chromatography, and different mass analysers. The Thermo Scientific platform used here lacks an ion mobility device. However, for the latter system such a device is available, and it would have been a minimal requirement to include it in such a comparative evaluation.*

Author's reply: We completely understand and agree with the concerns raised by all reviewers regarding the direct comparison between timsTOF and Orbitrap without ion mobility. As already stated in the reply to comment #2 it was never our intention to conduct a competitive comparison of devices from different vendors. Therefore, we believe that a direct benchmark testing between the timsTOF and a FAIMS coupled Orbitrap system would be pinning two systems of two vendors against each other and create a competitive environment and thus be understood as a sales recommendation. We would like to emphasize that this was not our intent and the scope of the manuscript.

Rather, our intention of the immunopeptidome analysis of paired samples on timsTOF and Orbitrap was to delineate the eligibility of TOF_{IMS} for HLA-presented peptide analysis, in particular to provide novel insights to and expansion of the immunopeptidome landscape of large-scale benign and malignant datasets. Therefore, we performed the alignment of paired

primary benign and malignant samples analyzed using TOF_{IMS} with the current state-of-the-field (FT-ICR) technology, applied in current immunopeptidome references which is as detailed in the reply to comment #1 the Orbitrap technology.

The intention of analyzing paired samples using TOF_{IMS} and FT-ICR technology was stated more clearly in the revised manuscript (please refer to line 40 - 43). Moreover, as also mentioned in reply to comment #1 and #2 in more detail the immunopeptidome analysis of paired samples on TOF_{IMS} and FT-ICR was now integrated in the method development section and has been significantly reduced and toned down with avoidance of any comparative statements and claims of superiority (please refer to lines 74 - 105 in the *TOF_{IMS} MS enables large-scale identification of naturally HLA-presented peptides* results section).

Comment #4: *It is unclear why the extended benign datasets are of value for cancer antigen discovery (which they clearly are), as authors highlight identification of neoantigens, which have been the result of tumour-specific mutation and therefore do not benefit from the additional analysis of benign tissues.*

Author's reply: We thank the reviewer for this comment and completely agree that benign reference data in immunopeptidomics are of central importance for the identification of tumor-associated antigens derived from non-mutant gene products. In recent years, the scientific community has mainly been interested in the discovery of neoantigens derived from tumor-specific mutation. However, only a minority of mutations at the DNA level is translated and naturally processed to HLA-presented neoepitopes targetable for T cells (Yadav *et al.* Nature 2014, Finn *et al.* Cold Spring Harb Perspect Biol 2018, Bassani-Sternberg *et al.* Nat Commun. 2016). Non-mutated tumor-associated antigens, arising through differential gene expression or protein processing in tumor cells, have been shown to play a central role for tumor immune surveillance and thus represent highly promising target candidates for immunotherapy, in particular for low-mutational burden entities (Godet *et al.* Clinical Cancer Research 2012, Schmitt *et al.* Blood 2008, Kowalewski *et al.* PNAS 2015, Tegeler *et al.* Cancers 2022, Marconato *et al.* Cancers 2022). To prove their tumor exclusivity, subtraction of extensive benign reference data is indispensable to enable safe clinical application.

Accordingly, as requested by the reviewer we rephrased and highlighted text passages in the introduction and discussion to further emphasize the importance of benign immunopeptidome references for tumor antigen discovery (please refer to lines 20 - 22 and 203 - 208).

Comment #5: *Therefore, I believe this manuscript is not currently suitable for publication in Nature Communications. I would suggest a removal of comparative data, and the publication of the highly relevant benign tissue data as an extension of the HLA ligand atlas first published by the authors in 2021.*

Author's reply: We thank the reviewer for the valuable suggestion how to improve the manuscript and the positive evaluation of our benign immunopeptidome dataset. Based on the reviewer's recommendations we completely restructured our manuscript in order to present more clearly the focus of the manuscript, which is (i) the introduction of an immunopeptidomics methods for TOF_{IMS}, (ii) the development of a novel benign_{TOFIMS} immunopeptidome dataset, and (iii) its' application in antigen target definition. The immunopeptidome analysis of paired samples on timsTOF and Orbitrap, which was a substantial part of the first manuscript version, was now integrated in the method development section, has been significantly reduced and toned down with avoidance of any comparative statements and claims of superiority. In addition, we removed any claims of superiority from the revised introduction and discussion section and included a detailed discussion on the benefits of IMS use in general in the discussion section of the manuscript. (please refer to line 191 - 200). Moreover, we have revised the naming of both devices to TOF_{IMS} and FT-ICR to focus more on the technical features instead of promoting a specific brand or instrument.

Our revised manuscript now emphasizes the methodological development and the biological relevance of its' usage in immunopeptidomics. To this end we have included multiple new analyses and experiments that are presented in the reconstructed figures. We have characterized the novel benign dataset by looking at tissue specific yields, peptide length analysis and peptide overlap of different tissues from the same donor (please refer to new Figure 3 and new Supplementary Figure 3). Furthermore, we compared in more detail how the novel peptides identified differ from already described benign-associated peptides (please refer to new Figure 4) by analyzing their biochemical properties, such as their amino acid composition and hydrophathy, their predicted immunogenicity, and their biological relevance by aligning KEGG pathway annotation and subcellular location of source proteins. In addition to the application of the benign dataset in refining the definition of tumor antigens (previously Figure 3, now please refer to new Figure 5 and new Supplementary Figure 4 in the revised manuscript), we provide information on the origin of peptides identified in the benign_{TOFIMS} immunopeptidome dataset that were described as tumor-exclusive in previous studies.

We would like to thank the reviewer again for the highly appreciated input on how to further improve our manuscript and hope that you are content with the additional experiments, data analyses, and revisions we made to the manuscript.

Reviewer #3 (Remarks to the Author):

T cell recognition of human leukocyte antigen (HLA) presented tumor associate peptides is central for cancer immune surveillance. This manuscript describes the use of timsTOF with ion mobility separation-based MS for immunopeptidomics profiling. Comparisons between MS data generated via IMS timsTOF versus Orbitrap were also performed. The authors found a significant gain of immunopeptide acquisitions through the timsTOF. The authors then analyzed MHCII peptidomes from up to 94 benign tissues and 4 primary malignant samples and identified 150,000 new peptides.

Author's reply: We thank the reviewer for the kind review and highly appreciate the input on how to improve our manuscript. Please find below a detailed point-to-point reply to the specific questions and issues raised.

Comment #1: *The authors should describe in more details how the timsTOF specifically allows for the gain of identifications compared to the Orbitraps. For example, are there any physiochemical properties unique to the immunopeptides that render them more challenging to be detected without IMS capability? How are detections hampered by Orbitrap alone? At present it is hard to figure out what to conclude from the comparison with the Orbitrap. It is clear different instruments have different strengths and the timsTOF is a faster machine. Some of the performance gains from the timsTOF data set might also have been reached using better separation and longer acquisition time on the Orbitrap.*

Author's reply: We appreciate the reviewer's request for a more detailed discussion on the underlying reason for the gain of identifications using IMS in the timsTOF system (referred to as TOF_{IMS} in the revised manuscript) and Orbitrap (referred to as FT-ICR in the revised manuscript). Although ion mobility has many advantages, such as improving prediction algorithms (Zeng *et al.* Nat Commun 2022), we believe that ion mobility itself is not the central factor for the gain in identifications observed here. Rather the increased/orthogonal separation and synchronized faster sequencing speed that comes with it leads to an improved sensitivity and thus higher peptide yields. Klaeger *et al.* Mol Cell Proteom. 2021 could show a peptide yield increase using the high-field asymmetric waveform ion mobility spectrometry (FAIMS) system and showed that fewer co-isolated peptide precursor ions were identified thus leading to fewer chimeric spectra and subsequently to more identifications based on an improved spectra annotation. This is in line with our observations for the TOF_{IMS} system, where up to 96% yield increase was observed. By adding the additional separation dimension, in particular in the trapped-ion-mobility setting, the relatively homogenous immunopeptide pool can be separated more efficiently and synchronized to the MS acquisition. In the revised manuscript we included a discussion of the underlying reasons for the gain of identifications (please refer to line 191 - 196).

To further illustrate this point, we analysed our dataset for peptides eluting in the same 10 seconds time frame (representing a typical elution peak with the nanoElute TOF_{IMS} system) and how many of these peptides overlap with their m/z. These peptides would potentially not be distinguishable in a conventional LC-MS setting as the precursor might not be picked for fractionation and MS/MS identification. We found that 0.2% (34 of 18,359 peptides) of peptides matched these criteria. This shows that not the actual ion mobility spectrometry is aiding in the majority of peptide identifications, and that it is more likely that the separation capacity added by IMS is responsible for the improved peptide separation and detection rather than the actual ion mobility itself.

As requested by the reviewer we further characterized the exclusive peptides identified in the immunopeptidomic data of using TOF_{IMS}, FT-ICR acquired data, as well as published benign data with regard to their physiochemical properties, comprising length, mass, hydrophathy and amino acid composition, of the peptides identified. Whereas, no relevant differences in terms of length, mass and amino acid compositions was found, we could show that that TOF_{IMS} exclusive peptides were significantly more hydrophobic and thus predicted to be more immunogenic (using IEDB's immunogenicity prediction tool) compared to published and FT-ICR acquired data (please refer to Figure 2 e/f, new Supplementary Figure 2 c – f, and new Fig. 4d – g).

To further address the request of the reviewer for a longer separation gradient of the FT-ICR system to improve peptide identifications we analyzed three different gradients: (i) the standard 90 min, (ii) 120 min separation and (iii) 150 min separation gradient, all three followed by a 30 min wash step. We found that increasing the gradient length did not result in an increase of peptide identifications, see Reviewer only Figure 1. This might be due to the low abundance of peptides in many immunopeptidomic samples. By increasing the gradient length, the intensity suffers through wider peaks and results in a yield drop. We have also observed this for the TOF_{IMS} liquid chromatography (please refer to Figure 1a) where the peptide yields dropped at a certain gradient length.

Reviewer only Figure 1 | Influence of liquid chromatography separation gradient length on HLA class I-presented peptide yields. Three gradient lengths (90, 120, 150 min) were tested using the same gradient conditions (2%-32% acetonitrile followed by a 30 min wash step) in an EoL cell line sample (7.5×10^5 cells).

Comment #2: *The technical or methodological novelty of the study is not clearly stated. Even though it is a relatively newer technology than the Orbitraps, the Bruker timsTOF Pro has been introduced since 2017, and it is widely utilized by the proteomics community nowadays, and so is not exactly a novel MS instrument in 2023. MHC-I immunopeptide data using a timsTOF instrument with PASEF have been demonstrated before (Feola et al. eLife 2022). The authors should cite these studies and better describe how the current approach differs.*

Author's reply: We highly apologize for not citing the work of Feola et al. eLife 2022, which was now included in the introduction and discussion section of the revised manuscript (please refer to lines 43 and 198). In Feola et al.'s manuscript a novel immunopeptidomic-based pipeline for the further development of their oncolytic adenovirus platform PeptiCRAd was described, applying timsTOF MS for HLA class I analysis in a single cell line.

In our study we present the detailed establishment and validation of a method for immunopeptidomics using the TOF_{IMS} system for the large-scale analysis of both HLA class I and HLA class II-presented peptides from primary samples and present its' application for the establishment of a large novel benign_{TOFIMS} immunopeptidome dataset and the refinement and *de novo* identification of tumor antigens.

Compared to Feola et al.'s our method injects less sample over a longer gradient with a larger mass window. During method development we found that increasing the m/z window drastically improved the peptide yields (from 2,995 HLA class I peptides to 4,085 HLA class I peptides in a JY sample, please refer to Figure 1d). Moreover, Feola et al. used a smaller $1/k_0$ (0.6 - 1.3 Vs/cm² compared to our 0.6 - 1.6Vs/cm²) window and only 3 PASEF scans compared to 6 scans in our approach. We found these factors to be contributing to improved peptide yields (2,249 HLA class I peptides compared to 3,014 HLA class I peptides, please refer to Figure 1b - h).

We included a discussion of the work of Feola *et al.* in the discussion section of the manuscript (please refer to line 43 and 198). Moreover, in response to this comment and based in the suggestions of the other reviewers, we have completely restructured our manuscript in order to present more clearly the focus of our manuscript, which is (i) the introduction of an immunopeptidomic methods for TOF_{IMS}, (ii) the development of a novel benign_{TOFIMS} immunopeptidome dataset, and (iii) its application in antigen target definition (line 45 and 50).

Comment #3: *The manuscript will benefit from additional analyses focusing more on the new peptides identified and their biological relevance instead of being heavily centered on the numbers of identification. Currently I feel the manuscript mostly describes the availability and depth of a data set without a lot of analysis on what it all means.*

Author's reply: We thank the reviewer for this valuable suggestion to improve our manuscript. We agree with the reviewer that the manuscript will benefit from additional analysis providing a more in-depth analysis of the immunopeptidome dataset. Therefore, we have included various new analyses and completely restructured the manuscript. The revised version includes a new Figure 3 and new Supplementary Data 3, that presents a detailed overview of peptide characteristics from the benign dataset. This comprises an overview of samples, an HLA allotype population analysis covered by the sample cohort, a sample specific peptide overview, a tissue specific peptide analysis, as well as an overlap analysis of peptides identified on multiple tissues from the same donor. Furthermore, the revised manuscript includes additional analysis of the novel identified peptides presented in the new Figure 4d - k. As requested the novel identified peptides were analyzed for their biological relevance compared to already published peptides from benign references by comparing source protein KEGG pathway annotation (please refer to Figure 4h and i) and the subcellular location of source proteins (please refer to Figure 4j and k). Of note, as already discussed in the response to comment #1, the newly identified peptides do not significantly differ from already described benign associated peptides in terms of biological properties and sources. This shows that the increased sensitivity leads to the identification of low abundant peptides rather than a novel kind of peptides. In addition, the application and biological relevance of the novel identified benign peptides is presented in a new Figure 5, where tumor-associated antigen (TAA) definition is refined by using the novel benign_{TOFIMS} peptides as well as an analysis showing the origin of the rejected TAA (please refer to new Supplementary Figure 4a). This underlines the importance of understanding the broad and complex HLA presentation on different tissues. Furthermore, we included a Supplementary Data 3 that contains detailed information on all sample in terms of identified peptides, PSMs and peptide source proteins.

Comment #4: *Are there any quantitative information that can be drawn from the MS data? Since the authors mentioned low abundance peptides at several places, some relative*

quantification could potentially offer some insights into the relative abundance of these peptides within individual and across the different samples. Are there significant variability among the different biological replicates? How does the reproducibility compare between the Orbitrap and timsTOF in terms of signal intensity across samples and also identification?

Author's reply: We thank the reviewer for the interest in the identification of low abundant peptides. The here performed discovery immunopeptidomics approach does not allow for an absolute quantification of HLA-presented peptides. Nevertheless, to address this point we included a semi-quantitative analysis of TOF_{IMS} data and FT-ICR data (please refer to new Supplementary Fig. 2i) where the reported area of peptides identified from the ten paired samples analysed on TOF_{IMS} and FT-ICR were compared. Peptides without a reported area were excluded from both datasets. The reported area per peptide was normalized for each sample (device-specific and sample-specific) and then ranked. The peptides were further marked as shared (when identified in both datasets) or TOF_{IMS}/ FT-ICR exclusive. Of the peptides ranked in the lower third, 41% were TOF_{IMS} exclusive compared to 20% FT-ICR exclusive. This new data was included in the revised result section in the manuscript (please refer to lines 92 - 93 and new Supplementary Figure 2i).

Of note, PEAKS does often not report an area for peptides of low abundance and these were excluded from the semiquantitative analysis mentioned above. Of note, 40% of TOF_{IMS} identified peptides (78% of these were exclusive) compared to 3% of FT-ICR identified peptides did not have a reported area (yet a confident PSM and thus remain included in the dataset). This observation further supports the increased identification of low abundant peptides using TOF_{IMS}.

Comment #5: *The lack of personalized genome specific protein sequence database (with SAAV or tumor mutations) for most of the samples (except the HNSCC samples) is in my view a technical weakness that limits the utility of the data set. It perhaps also raises some question about identification fidelity when some mutant peptides are probably present that cannot be identified with the database used. For the HNSCC mutations it was not clearly explained whether the exome data were from matching samples as the immunopeptidome data.*

Author's reply: We apologize for not clearly stating that the exome sequencing of the HNSCC sample is of the same tissue/ tumor origin as the immunopeptidome analysis and have therefore rephrased the method section (please refer to line 297 - 301).

We further acknowledge the limitation of the described benign immunopeptidome dataset due to the lack of sample specific sequencing, which indeed does not allow to identify SAAV in the benign immunopeptidome dataset as well as SAAV and/ or mutant peptides for the CLL and RCC samples. Considering the cohort's sample size and the controversial interplay between

transcriptome, proteome and immunopeptidome these analyses were beyond the scope of this work. We added this limitation of our work in the discussion section of the revised manuscript (please refer to line 236 - 238).

Comment #6: *The results section stated that the current data set rejected up to 40% of previously published OvCa TAAs. The authors may want to expand on what kind of peptides were rejected, or whether the retained peptides now constitute true TAA. It is already recognized that neoepitopes provide better TAAs for immunotherapy (one might assume the current data set demonstrates rather the danger of using non-mutated peptides as TSA/TAA instead since if sensitivity is unlimited then presumably every protein/peptide might be expected to be loaded on an MHC).*

Author's reply: We thank the author for raising this point. Indeed, mutation-derived neoantigens represent promising target for immunotherapeutic approaches, and have already demonstrated immunogenicity and first clinical efficacy in particular in high-mutational burden tumor entities. However, only a minority of mutations at the DNA level is translated and naturally processed to HLA-presented neoepitopes targetable for T cells (Yadav *et al.* Nature 2014, Finn *et al.* Cold Spring Harb Perspect Biol 2018, Bassani-Sternberg *et al.* Nat Commun. 2016). It has been shown that for primary human tumor samples the frequency of naturally presented neoantigens identified by MS-based immunopeptidomics ranged from 0.00% to 1.25% per sample (Bassani-Sternberg *et al.* Nat Commun 2016), suggesting a very low abundance of mutation-derived neoepitopes. Within this work we could show that with the increased sensitivity of TOF_{IMS}-based MS it was possible to detect naturally presented neoepitopes in a primary HNSCC sample.

Nevertheless, in particular for low-mutational burden tumor entities alternative antigen targets are required for immunotherapy. Here non-mutated tumor-associated antigens, arising through differential gene expression or protein processing in tumor cells, might supplement or even substitute neoepitope targeting. To prove tumor exclusivity for these antigens large scale benign immunopeptidome control references are required. Despite the constant growth of these databases, the landscape of the whole benign immunopeptidome is still not covered and it is unclear when a saturation of peptide identifications is/will be reached. This calls for a continuous re-evaluation of antigen targets and expansion of benign immunopeptidome databases as both provided in this manuscript.

Moreover, selection of optimal antigen targets is never solely based on the natural and tumor exclusive presentation but requires prove of immunogenicity based on the detection of pre-existing immune responses in tumor patients and/or *de novo* induction of peptide-specific immune responses in naïve T cells and subsequent efficacy evaluation in *in vivo* studies.

We included a more detailed discussion on the different kinds of tumor antigens and the prerequisite of benign reference immunopeptidomes for the validation of non-mutated tumor antigens in the revised discussion section of the manuscript (please refer to line 201 - 208).

Furthermore, as requested by the reviewer we provide more details on the rejected peptides and their representation on specific benign tissues. Of note, whereas the rejected peptides from OvCa were identified on various different solid tissues and donors, the rejected CLL and CML peptides were primarily found on PBMCs and bone marrow from healthy volunteers (please refer to new Supplementary Figure 4a and results section lines 155 - 157).

Comment #7: *Since MHC peptides are usually considered challenging to identify, the authors may want to compare different search engines and post-processing workflows, and also consider whether 1% FDR cut-off as opposed to spectrum- or peptide- specific error probability is sufficiently stringent.*

Author's reply: We thank the reviewer for raising this point. We agree that different search engines and FDR calculations have an influence on peptide identification. We appreciate the reviewer's interest in further benchmarking and optimizing bioinformatic immunopeptidomic workflows, however this goes beyond the scope of the manuscript, which was to (i) implement an immunopeptidomic method development for TOF_{IMS}, (ii) the generation of a novel benign immunopeptidome dataset, and (iii) its application in antigen target definition. Moreover, so far only a few annotation tools are able to analyze both raw data formats (.d as well as .raw), in particular without prior conversion to a different file format.

We further agree that a benchmarking study comparing different filter criteria would be of significant interest, however, is beyond the scope of this study. We believe the applied FDR cut-off of 1% is appropriate for the identification of valid HLA-presented peptides based on previous work by us and others applying the 1% FDR or even higher cut offs (e.g. up to 5% FDR) with subsequent synthetic peptide spectrum validation (Klaeger *et al*/Mol Cell Proteomics 2021, Jaeger *et al*. Nature 2022, Ternette *et al*. Eur J Immunol 2016, Bilich *et al*/Blood 2019). The cut-off filter of 1% FDR was complemented by filtering and annotation of identified peptide sequences to the HLA allotype of the respective sample using HLA binding algorithms (please refer to line 309 - 312). This provides a further layer of validation and enables the exclusion false positive peptide sequences not matching the respective HLA allotype binding motif.

We would like to thank the reviewer again for the highly appreciated input on how to further improve our manuscript and hope that you are content with the additional experiments, data analyses, and revisions we made to the manuscript.

Reviewer #4 (Remarks to the Author):

In their manuscript entitled “timsTOF mass spectrometry-based immunopeptidomics refines tumor antigen identification”, Gravel et al. compare the performance of a timsTOF mass spectrometer-based workflow to an Orbitrap-based workflow. The authors apply the developed TimsTOF method towards the characterization of the MHC class I and MHC class II ligandomes of 94 benign tissues and show principal applicability of the workflow towards neoantigen discovery.

The manuscript is well written and figures are of high quality. Presentation of the data is clear, but some results are shown at a very highly aggregated level that is in some cases potentially prone to misinterpretation. Overall, the work is of high significance to the field, but some analyses need to be explained in more detail and supported by supplementary figures and tables to enable them to be easily reproduced by other researchers.

Author reply: We thank the reviewer for the kind review and highly appreciate the input on how to improve the manuscript. Please find below a detailed point-to-point response, describing how we addressed the specific comments and concerns.

Comment #1: *Claims of novelty should be toned down, as the application of timsTOF to immunopeptidomics is not completely novel (DOI: 10.3389/fimmu.2022.932252), Casasola-LaMacchia et al. described the identification of 12.000 HLA-II associated peptides from dendritic cells.*

Author's reply: We apologize for not mentioning the work of Casasola-LaMacchia et al. Front. Immunol. 2022. In the revised manuscript, as requested by the reviewer, we toned down any claims of novelty and included to our knowledge all references describing immunopeptidome studies using timsTOF (further referred to as TOF_{IMS} in the revised manuscript) devices (please refer to line 41).

Comment #2: *Figure 1d: This is described as the distributions of peptide m/z from a JY-2 sample, but the total number of peptides in this panel seems to be 10x higher than in the other panels.*

Author's reply: We apologize for the confusing presentation of data. The graphs of Figure 1 display data from two different concentrations of JY-cell samples (JY 1 equivalent to 1.2×10^7 cells and JY 2 equivalent to 8×10^6 cells). Figure 1a displays data from both JY 1 and JY 2, whereas Figure 1b displays only data from JY 2, and Figures 1c - f show the results of a JY 1 sample measurement. To present the data more clearly, we have changed Figures 1c – f (Figures 1c, e, f in the revised manuscript) now presenting data from a JY 2 sample.

Comment #3: The authors compare the timsTOF instrument to an Orbitrap Fusion Lumos platform. Klaeger et al. previously showed a beneficial effect of FAIMS ion mobility for Orbitrap instruments (DOI: 10.1016/j.mcpro.2021.100133), which significantly enhanced sensitivity. To make a fair comparison, the authors ideally should compare timsTOF vs. a FAIMS-equipped Orbitrap platform on selected samples.

Author's reply: We thank the reviewer for making this point. We completely understand and agree with the concerns raised by all reviewers regarding the direct comparison between timsTOF and Orbitrap without ion mobility. Our intention of the immunopeptidome analysis of paired samples on timsTOF and Orbitrap was to delineate the eligibility of TOF_{IMS} for HLA-presented peptide analysis, in particular to provide novel insights to and expansion of the immunopeptidome landscape of large-scale benign and malignant datasets. Therefore, we performed the alignment of paired primary benign and malignant samples analyzed using timsTOF with the current state-of-the-field (FT-ICR) technology, applied in current immunopeptidome references which is as shown in a PubMed search (see Reviewer only Figure 1) the Orbitrap technology.

Reviewer only Figure 1 | MS devices used in the field of immunopeptidomics. Percentage of publications (including benign tissue immunopeptidome databases) on PubMed using different MS devices from 2018 - 2023 (April). The absolute number of publications is indicated above the bar.

The intention of analyzing paired samples using TOF_{IMS} and FT-ICR technology was stated more clearly in the revised manuscript (please refer to line 44 - 46). Moreover, the immunopeptidome analysis of paired samples on timsTOF and Orbitrap was now integrated in the method development section and has been significantly reduced and toned down with avoidance of any comparative statements and claims of superiority (please refer to the abstract, results and discussion sections). Moreover, we have revised the naming of both devices to TOF_{IMS} and FT-ICR to focus more on the technical features instead of promoting a specific brand or instrument.

We believe that a direct benchmark testing between timsTOF and a FAIMS coupled Orbitrap system would be pinning two systems of two vendors against each other and create a competitive environment and thus be understood as a sales recommendation. We would like to emphasize that this was not our intent and the scope of the manuscript.

Comment #4: *Figure 2a: It is not clear how these numbers (ligands per tissue) were derived. Are these identifications pooled from different patients? What is the number of ligands found in each patient sample? How reproducible are the results, i.e. what is the overlap between technical replicates in the tissue samples? The overall ligand numbers across 94 tissues reported are quite impressive, but the authors also need to show what would be expected from the analysis of individual tissues/samples, ideally on technical replicate level, as a supplementary figure and provide a supplementary table containing the numbers of peptides identified in each of the samples.*

Author's reply: We apologies for the confusing presentation of data. The numbers in Figure 2a and b (changed from Extended Data 2 in the revised manuscript) are shown per individual tissue sample. To present the data more clearly, we indicated the sample ID (same as in Supplementary Data 2) and labelled the corresponding samples in Figure 2a and b. Furthermore, we rephrased the figure legends to clearly state that the numbers reported stem from individual tissue samples (please refer to line 402). In addition, we added a new Supplementary Data 3 showing the peptide yields for each sample (including the ten comparative and the whole benign_{TOFIMS} data set presented in the new Figure 3). Moreover, the method description of HLA ligand isolation was rephrased in the revised manuscript stating more clearly that ten samples were measured using both MS devices and the 94 samples included in the benign_{TOFIMS} were solely acquired using the TOF_{IMS} system (please refer to lines 261 - 268).

As the samples included in this manuscript are patient derived primary samples, no biological replicates could be measured. However, each sample was measured in 3 technical replicates. To provide data on the technical reproducibility Supplementary Figure 2 was expanded and the new Figure 3 contains information on the reproducibility within the benign_{TOFIMS} dataset (please refer to Figure 3f, line 118 - 120). The mean percentage of peptides found in all three technical replicates of the benign_{TOFIMS} dataset was 54% for both HLA class I and II-restricted peptides (range 32% to 63% and 35% to 64%, respectively), and 20% for HLA class I and II found in two replicates (range 17% to 32% and 16% to 33%). A mean of 26% and 27% of HLA class I and II-presented peptides (range 7% to 51% and 6% to 41%, respectively) were found in only one technical replicate across the whole benign_{TOFIMS}. Overlap analysis for the technical replicates of the ten samples measured on both systems, showed a mean percentage of 70% and 63% HLA class I and II-presented peptides, respectively overlapping in all three replicates with the FT-ICR system compared to 59% and 58% for peptides identified using TOF_{IMS} (new Supplementary Figure 2j, please refer to lines 93 - 96)

Comment #5: *Figure 3c: The authors seem to compare the results here on a very global level - it is not at all clear from which sample(s) the data are derived. Analyzing the data on Proteomexchange indicates that some timsTOF datasets have fewer than 500 identifications.*

Author's reply: We apologize for the unclear presentation of the data. We revised Figure 3c now showing a scatterplot representing each sample included in the corresponding databases and the mean value of peptide identifications in the cohort. The corresponding figure legend was adapted accordingly (please refer to lines 435 - 437). In addition, we included a new Supplementary Data 3 that provides all peptide yields, PSM and source proteins identified for each sample in the benign_{TOFIMS} data set in addition to a new Figure 3 that illustrates peptide yields as well as the coverage of the world population by the HLA allotypes within in the database. It further includes tissue peptide yields, length distribution, the technical overlap within samples, as well as the tissue overlap for four donors with multiple tissues within the benign data set in form of chord plots. In the revised manuscript we have referred and described this new information in the results section (please refer to lines 112 - 127).

Comment #6: *What is the overlap in presented ligands between different tissues from the same patient? This analysis should be easy to derive from the data and could provide important insights regarding tissue-specificity of antigen processing.*

Author's reply: We appreciate the reviewer's suggestion for further analysis of the benign immunopeptidome dataset including insights into the tissue-specificity of peptides. As mentioned in the previous comment (please refer to comment #5) we have included additional analysis of the presented benign_{TOFIMS} dataset in Figure 3, which also describes the overlap of HLA ligands between the different tissue samples of four exemplary donors with multiple tissues (n = 11 - 20, please refer to Figure 3h, Supplementary Figure 3 and lines 121 - 127). As already described before (Marcu et al. Mol Cell Prot, 2021), we found that although many HLA ligands are found in multiple tissues, approximately 50% of HLA class I and 70% of HLA class II-presented peptides were of only one tissue origin, indicating a tissue specificity and further underscoring the importance of large-scale and diverse benign reference databases in order to ensure proper tumor antigen identification.

Minor comments:

Comment #7: *The specific coordinates of isolation polygons (V2/V3/V4) used need to be defined to enable other groups to reproduce the findings.*

Author's reply: We apologize for not precisely describing the specific coordinates of the isolation polygons in our manuscript. In the revised version the Supplementary Data 1 has been adapted to include screenshots of all four (V1, V2, V3, V4) polygons to ease reproducibility.

Comment #8: Line 176-182 this statement should be modified, as it reads somewhat like a Bruker marketing brochure.

Author's reply: Again, we would like to apologize for highlighting a specific manufacturer. As mentioned above (please refer to comment #1) we adapted the manuscript in order to down tone claims of superiority and not further pin both companies (Thermo Fisher and Bruker) against each other. The specifically mentioned statement (line 176-182, now line 191 - 196) was adapted as follows:

"In line with improvements reported for other omics technologies, as well as with FAIMS technology for immunopeptidomics, TOF_{IMS} methodology enabled high sensitivity and fast track peptide identification. IMS provides a new dimension of separation with the CCS value as an additional peptide property, which has been suggested to improve statistical confidence in peptide identification. TOF_{IMS} efficacy relies on releasing ions according to their ion mobility synchronized to the mass analysis via TOF, resulting in a high speed and sensitive detection of HLA-presented peptides."

Comment #9: According to the search results deposited on ProteomeXchange, some peptides were not mapped to a source protein, or the respective identification is not provided. Were these peptides counted as identified? Please explain.

Author's reply: We thank the reviewer for this careful observation and agree that not all peptides were mapped to a source protein. When processing proteomics/ immunopeptidomics data with PEAKS studio, the software exports the top protein candidates per peptide. Since in bottom-up proteomics it is important to set appropriate peptide-protein mapping filters and criteria, PEAKS allows to select the "TOP" proteins as well as "ALL" proteins. Per default the "TOP" proteins are selected as protein annotation needs to carefully be considered in proteomics (Zhang *et al.* Chem. Rev. 2013). In immunopeptidomics however we are less interested from which of possible proteins a peptide is derived from. Since the protein analysis are limited in our case, the uploaded files were exported with the setting "TOP" and thus some peptides lack protein annotation. To address this issue, we reuploaded the processed files to the *ProteomeXchange* after reexporting with the "ALL" proteins setting.

Comment #10: Some spectra seem to be double annotated: e.g. in file 210831_NHG_benign_UDN04_Bladder_Tue39L243_20%_3RepsCoProcessed_peptide several precursor IDs are annotated with two different peptide sequences that differ only by a Leucine/Isoleucine replacement. This seems to artificially inflate the numbers reported.

Author's reply: We thank the reviewer for pointing out this issue. Indeed, during the spectral annotation, the used software PEAKS Studio does not distinguish between leucine/ isoleucine. If both sequences are included in the FASTA the software will include both in the resulting peptide list. In MS leucine and isoleucine are not differentiable due to their enantiomer status,

so far only Sciex ZenoTOF system is able to distinguish between them by using electron activated dissociation.

In the mentioned file *210831_NHG_benign_UDN04_Bladder_Tue39L243_20%_3RepsCo Processed_peptide* 5 of 566 precursors, <1%, were double annotated. We agree that this artificially inflates the peptide numbers reported, however this is the case for all data included in this manuscript, as all raw data was processed using the PEAKS Studio software. We included this limitation in discussion section of the revised manuscript (please refer to line 236 - 238).

Comment #11: *In the PEAKS reports, a significant proportion of peptides (20-30%) has a reported area of 0 in all replicates. Were these counted as identified? Please clarify / explain.*

Author's reply: We appreciate the thorough review of our uploaded data. Indeed, there are peptides in the PEAKS reports with a peptide area (or intensity mode when activated) of 0. The Bioinformatics team of the PEAKS software provides the following explanation for this issue: To get a reported area or intensity for a peptide a feature needs to be detected as a series of corresponding m/z values with similar RT range and MS1 intensities. In cases where a peptide is sporadically detected due to its low abundance or high matrix interference at its elution time point, a feature cannot be computed mathematically. Although a feature cannot be derived, and thus no area reported, a confident PSM can be made based on the MS2 scan that is found corresponding to the precursor. In these situations, the feature is defined as 0 and no area under the curve or intensity is provided as there is no elution profile in terms of MS1 scans. Since the PSM scoring however is done with enough confidence, we include these identifications as true identifications, however these peptides cannot be used for any (semi)quantitative analyses. Of note, the FT-ICR runs produce less peptides with a reported area of 0 compared to the TOF_{IMS} acquired data, which aligns with the observation that the TOF_{IMS} is able to identify more low abundant peptides.

Comment #12: *How do the authors treat peptides that do not match the expected length distribution of MHC ligands? Do the authors e.g. count 9-11 mers as identified ligands for MHC class II or 15+ mers as ligands for MHC I? Please clarify and discuss and provide the decision criteria.*

Author's reply: We appreciate the reviewer's interest in the complexity of HLA-presented peptides' lengths. As mentioned in the method section (please refer to lines 308 - 309) we included 8 - 16-mers and 8 - 30-mers for HLA class I- and HLA class II-restricted peptides, respectively. According to the current field there is no clear guide which peptide lengths should be in- or excluded in immunopeptidome analyses. The included lengths in literature range from 8 - 12-mers (Marcu *et al* J Immunother 2021) up to 7 - 25-mers (Parker *et al*. Mol Cell

Proteomics 2021) for HLA class I-presented peptides and 8 - 25-mers (Marcu *et al.* J Immunother Cancer 2021), 12 - 25-mers (Solleder et al iScience 2022) up to 9 - 29-mers (van Balen *et al.* J Immunol 2020) for HLA class II-presented peptides. As this manuscript also provides a benign reference dataset for future analyses, our intention was to cover a large range of peptides. Thus, with the applied length cut-offs 87% and 95% of all identified HLA class I/ II-presented peptides, respectively, were included in the benign_{TOFIMS} dataset (see Reviewer only Figure 2).

Reviewer only Figure 2 | HLA-presented peptide length distribution. Relative abundance of identified HLA class I (grey, n = 92) and HLA class II- (red, n = 94) presented peptide lengths from benign_{TOFIMS} dataset. The marked areas represent the peptides included in the final data set.

Due to the biochemical properties of HLA molecules, we interpret shorter peptides, i.e. 7-mers, are a result of cleavage products or similar processes. Too long products are to be seen as contaminations.

Comment #13: *The cycle time for the 8 PASEF method indicated in Supplementary Table 2 seems to be incorrect.*

Author's reply: We apologize for this incorrectness. The cycle time for the 8 PASEF scan method was corrected to 2.27 s in the revised manuscript.

We would like to thank the reviewer again for the highly appreciated input on how to further improve our manuscript and hope that you are content with the additional experiments, data analyses, and revisions we made to the manuscript.

REVIEWER COMMENTS

Reviewer #1 (Remarks to the Author):

I thank you for doing all you can to address the many points raised by other reviewers, including my comments, along with your thoughtful responses.

However, with regard to proving technical superiority over existing technology, which most reviewers pointed out, the authors surprisingly gave up on that.

They say that they decided to focus only on the biological importance, but they have not actually identified epitopes that could be targets for new cancer immunotherapies, nor have they proven that they are effective in the search for new targets.

This is hardly a demonstration of medical breakthrough.

Finally, as long as the technological breakthrough cannot be demonstrated, I think that it can no longer be said to be extremely significant novel and progressive enough to be published in this journal.

Reviewer #2 (Remarks to the Author):

The authors have addressed some of my concerns and have rewritten parts of the manuscript.

I am very glad to see more emphasis on the benign datasets, which are, as I stated before, clearly very valuable and will provide the community with an important extension of the current HLA Atlas.

It is widely known that the TIMS-TOF series from Bruker is highly suitable for immunopeptidomics acquisition. In my view nothing has changed on the lack of novelty (and the inappropriateness) of the direct instrument comparison, which I suggest to remove entirely from the manuscript. Orbitraps are not FT-ICRs, but are defined as their own type of MS analyser, so in addition, the data is incorrectly labeled now.

Reviewer #3 (Remarks to the Author):

Thank you for the responses. I have no further comments.

Reviewer #4 (Remarks to the Author):

The authors present a significantly revised version of their manuscript, now focusing primarily on the results of the TOF-IMS platform.

The authors have addressed the reviewer comments in a satisfactory fashion. The manuscript has thereby significantly increased in presentation quality.

While the implementation of an already (at least in terms of the major parameters) established workflow for immunopeptidomics on the timsTOF platform carries somewhat limited novelty, the significant extension of the existing benign tissue HLA repository constitutes the major scientific finding with significant benefit for the community.

Minor comments:

1. Figure 3h – while visually nice looking, I find it very difficult to extract any useful information from the chord diagrams.
2. Line 42: The instrument type naming convention chosen by the authors is not correct and must be

revised. FT-ICR typically refers to an ion cyclotron resonance- based mass spectrometer. While both FT-ICR and Orbitrap-based instruments use fourier-transform to generate mass spectra, ICR uses a strong magnetic field to trap ions - typically generated by a helium-cooled superconducting electromagnet. In an Orbitrap, ions are trapped using an electrostatic axially harmonic orbital trap. (see excellent review by Alexander Makarov DOI: 10.1002/mas.21549)

3. Line 42/ 162: I would disagree that "orbitrap" is "conventional", as historically, the first immunopeptidomics analyses have been done using TOF technology. Maybe rephrase to "using Orbitrap-MS"

Reviewer #1 (Remarks to the Author):

Comment #1: *I thank you for doing all you can to address the many points raised by other reviewers, including my comments, along with your thoughtful responses. However, with regard to proving technical superiority over existing technology, which most reviewers pointed out, the authors surprisingly gave up on that.*

Author's reply: We thank the reviewer for acknowledging the revisions made to the manuscript. Concerning the technical superiority over existing technology our intention was to show how the TOF_{IMS} technology will benefit the field of immunopeptidomics. However, several of the reviewers understood the work as a performance review between two vendors and thus as a sales pitch, which was never our intention. As suggested by the reviewers we have largely reduced comparative data of with Orbitrap and in this second round of revisions addressing the suggestions of the editor and the reviewers completely moved this alignment to the supplement (please refer to Supplementary Fig 2 in the revised manuscript).

Comment #2: *They say that they decided to focus only on the biological importance, but they have not actually identified epitopes that could be targets for new cancer immunotherapies, nor have they proven that they are effective in the search for new targets. This is hardly a demonstration of medical breakthrough.*

Author's reply: To further emphasize the biological importance and relevance of our work in particular of the benign_{TOFIMS} for the characterization of novel tumor-associated antigens we performed a TOF_{IMS}-based immunopeptidome analysis of a cohort of CLL patients (n = 22) and a subsequent comparative immunopeptidome alignment for the definition of high-frequent CLL-associated peptides. The HLA allotypes include in the CLL cohort were comparable to the allotypes included in published benign datasets as well as the benign_{TOFIMS} dataset (Supplementary Fig 4a, b, c). A median of 11,706 HLA class I ligands (range 5,976 to 18,115) and 7,833 HLA class II-presented peptides (range 2,208 to 10,484) were identified per sample (Supplementary Fig 4d and Supplementary Data 3). In total 121,871 unique HLA class I ligands and 86,785 HLA class II-presented peptides were identified from the CLL cohort. Comparative immunopeptidome profiling was performed with published benign datasets^{8, 34} and resulted in 68% of HLA class I ligands (83,010 peptides, Fig. 4e) and 72% of HLA class II-presented peptides (62,224 peptides, Fig. 4f) presented CLL-exclusive. A further alignment of these CLL-exclusive HLA class I and class II peptides using the novel benign_{TOFIMS} dataset revealed (Fig. 4 e, f) 46% of CLL HLA class I ligands (55,791 peptides) and 68% of CLL HLA class II peptides (49,147 peptides) CLL-exclusive, thus annotating an additional 27,219 HLA class I ligands and 13,077 HLA class II-presented peptides as benign compared to previously published data. 727 HLA class I and 1,556 HLA class II CLL-exclusive peptides showed a broad presentation within the CLL cohort with a frequency above 20% (up to 59% for HLA

class I and 77% for HLA class II). Allotype specific peptide alignment of the most abundant HLA allotypes within the cohort (HLA-A*02, HLA-B*35 and HLA-C*07) even revealed CLL-exclusive peptides with presentation in up to 100% of HLA-matching samples (HLA-A*02 up to 100%, HLA-B*35 up to 100% and HLA-C*07 up to 50%) representing highly promising, broadly applicable antigen targets for immunotherapeutic approaches. Of note, 98% of these highly promising, broadly applicable antigen targets for immunotherapeutic approaches have never been described in previous large cohort CLL immunopeptidome studies (Kowalewski *et al* 2015 PNAS, Nelde *et al* 2021 Front. Immunol).

Within this proof of concept malignant study we could prove the biological importance of TOF_{IMS}-based immunopeptidomics and of the novel benign_{TOFIMS} dataset (please refer to lines 154 - 175 in the result section as well as Figure 4e and f and Supplementary Fig 4).

Comment #3: *Finally, as long as the technological breakthrough cannot be demonstrated, I think that it can no longer be said to be extremely significant novel and progressive enough to be published in this journal.*

Author's reply: We disagree with the reviewer in this point. We believe that our work is of central relevance for the field.

Immunopeptidomics is of central importance to elucidate the antigenic landscape of HLA-presented peptides, which in recent years have gained increasing relevance in the development of immunotherapies for the treatment of various disorders, comprising cancer, autoimmune and infectious diseases (Chong *et al*. Nat. Biotechnol. 2022). Thus, the intention of this manuscript was to show how the advance of more sensitive MS technology can be translated to the field of immunopeptidomics. Beside the development and characterization of a TOF_{IMS} method for the application in immunopeptidomics, this work provides an enormous pool of newly characterized peptides from benign tissue origin as (benign_{TOFIMS}) a comparative resource for future studies in the field. With this currently available reference databases were expanded by more than 50%, suggesting the benign_{TOFIMS} database as novel state-of-the-art benign reference for future selection and validation of non-mutated TAAs, which we now proved in a first malignant study (as described in more detail in the reply to comment#2).

Within the revised manuscript we pointed more clearly on the novelty of these findings (line 174 - 175 and 218 - 229).

Reviewer #2 (Remarks to the Author):

Comment #1: *The authors have addressed some of my concerns and have rewritten parts of the manuscript. I am very glad to see more emphasis on the benign datasets, which are, as I stated before, clearly very valuable and will provide the community with an important extension of the current HLA Atlas.*

Author's reply: We thank the reviewer for the kind assessment of the changes made to our revised manuscript and for recognizing the significance of the extension of the benign HLA data.

Comment #2: *It is widely known that the TIMS-TOF series from Bruker is highly suitable for immunopeptidomics acquisition. In my view nothing has changed on the lack of novelty (and the inappropriateness) of the direct instrument comparison, which I suggest to remove entirely from the manuscript. Orbitraps are not FT-ICRs, but are defined as their own type of MS analyser, so in addition, the data is incorrectly labeled now.*

Author's reply: We apologize and agree with the reviewer's assessment that the term FT-ICR is not the correct terminology for Orbitraps. In our revised manuscript we intentionally chose different naming for the instrument in order to avoid any emphasis on the competition between two vendors (Bruker and Thermo Fisher) and the proprietary aspects tied to the instrument's manufacturers. However, we agree with the reviewer that this is not correct and apologize for this mislabelling. We have re-changed the naming back to Orbitrap in the revised manuscript. Further as suggested by you and the editor have moved the former Fig 2 including the comparison of TOF_{IMS} with Orbitrap to the supplementary material (please refer to Supplementary Fig 2 in the revised manuscript).

To further emphasize the biological importance and novelty of our work in particular of the benign_{TOFIMS} for the characterization of novel tumor-associated antigens we performed a TOF_{IMS}-based immunopeptidome analysis of a cohort of CLL patients (n = 22) and a subsequent comparative immunopeptidome alignment for the definition of high-frequent CLL-associated peptides. The HLA allotypes include in the CLL cohort were comparable to the allotypes included in published benign datasets as well as the benign_{TOFIMS} dataset (Supplementary Fig 4a, b, c). A median of 11,706 HLA class I ligands (range 5,976 to 18,115) and 7,833 HLA class II-presented peptides (range 2,208 to 10,484) were identified per sample (Supplementary Fig 4d and Supplementary Data 3). In total 121,871 unique HLA class I ligands and 86,785 HLA class II-presented peptides were identified from the CLL cohort. Comparative immunopeptidome profiling was performed with published benign datasets^{8, 34} and resulted in 68% of HLA class I ligands (83,010 peptides, Fig. 4e) and 72% of HLA class II-presented peptides (62,224 peptides, Fig. 4f) presented CLL-exclusive. A further alignment of these CLL-

exclusive HLA class I and class II peptides using the novel benign_{TOFIMS} dataset revealed (Fig. 4 e, f). 46% of CLL HLA class I ligands (55,791 peptides) and 68% of CLL HLA class II peptides (49,147 peptides) CLL-exclusive, thus annotating an additional 27,219 HLA class I ligands and 13,077 HLA class II-presented peptides as benign compared to previously published data. 727 HLA class I and 1,556 HLA class II CLL-exclusive peptides showed a broad presentation within the CLL cohort with a frequency above 20% (up to 59% for HLA class I and 77% for HLA class II). Allotype specific peptide alignment of the most abundant HLA allotypes within the cohort (HLA-A*02, HLA-B*35 and HLA-C*07) even revealed CLL-exclusive peptides with presentation in up to 100% of HLA-matching samples (HLA-A*02 up to 100%, HLA-B*35 up to 100% and HLA-C*07 up to 50%) representing highly promising, broadly applicable antigen targets for immunotherapeutic approaches. Of note, 98% of these highly promising, broadly applicable antigen targets for immunotherapeutic approaches have never been described in previous large cohort CLL immunopeptidome studies (Kowalewski *et al* 2015 PNAS, Nelde *et al* 2021 Front. Immunol)

Within this proof of concept malignant study we could prove the biological importance of TOF_{IMS}-based immunopeptidomics and of the novel benign_{TOFIMS} dataset (please refer to lines 154 - 175 and 218 - 229 in the result section as well as Figure 4e and f and Supplementary Fig 4).

Reviewer #3 (Remarks to the Author):

Thank you for the responses. I have no further comments.

Author's reply: We thank the reviewer very much for the kind review and highly appreciated the positive evaluation of our revised manuscript.

Reviewer #4 (Remarks to the Author):

Comment #1: *The authors present a significantly revised version of their manuscript, now focusing primarily on the results of the TOF-IMS platform. The authors have addressed the reviewer comments in a satisfactorily fashion. The manuscript has thereby significantly increased in presentation quality. While the implementation of an already (at least in terms of the major parameters) established workflow for immunopeptidomics on the timsTOF platform carries somewhat limited novelty, the significant extension of the existing benign tissue HLA repository constitutes the major scientific finding with significant benefit for the community.*

Author's reply: We thank the reviewer for the kind assessment recognizing the importance of the presented benign HLA repository and its analysis.

Minor comment #1: *Figure 3h – while visually nice looking, I find it very difficult to extract any useful information from the chord diagrams.*

Author's reply: We apologize for the confusing presentation of the peptide overlap between tissues from the same donor. To allow the for an easier interpretation of the data presented, we have modified the plots to include a bar chart that illustrates the percentage of peptide overlap between the different number of tissues within one donor (please refer to Fig. 2h and i).

Minor comment #2: *Line 42: The instrument type naming convention chosen by the authors is not correct and must be revised. FT-ICR typically refers to an ion cyclotron resonance- based mass spectrometer. While both FT-ICR and Orbitrap-based instruments use Fourier-transform to generate mass spectra, ICR uses a strong magnetic field to trap ions - typically generated by a helium-cooled superconducting electromagnet. In an Orbitrap, ions are trapped using an electrostatic axially harmonic orbital trap. (see excellent review by Alexander Makarov DOI: 10.1002/mas.21549).*

Author's reply: We apologize and agree with the reviewer's assessment that the term FT-ICR is not the correct terminology for Orbitraps. We appreciate your clarification and agree with you on the key differences between FT-ICR and Orbitrap-based instruments. In our revised manuscript we intentionally chose different naming for the instrument in order to avoid any emphasis on the competition between two vendors (Bruker and Thermo Fisher) and the proprietary aspects tied to the instrument's manufacturers. However, we agree with the reviewer that this is not correct and apologize for this mislabelling. Thus, we have re-changed the naming back to Orbitrap in the revised manuscript.

Minor comment #3: *Line 42/ 162: I would disagree that "orbitrap" is "conventional", as historically, the first immunopeptidomics analyses have been done using TOF technology. Maybe rephrase to "using Orbitrap-MS".*

Author's reply: As suggested by the reviewer we have changed the corresponding phrases in the revised manuscript (please refer to line 42 and 145).

REVIEWERS' COMMENTS

Reviewer #2 (Remarks to the Author):

The authors have answered my comments by highlighting the biological significance of their work, which I have been in complete agreement with and believe is very suitable for publication in Nature Communications. My concerns were exclusively related to the instrument performance comparison and (now) Supp. Fig. 2/3, which has no real novelty nor value to the readership in my view. However, since this is now a minor point in an otherwise excellent manuscript, I suggest acceptance without further changes.